# Formation and fate of freshwater on an ice floe in the Central Arctic

Madison M. Smith[1], Niels Fuchs[2], Evgenii Salganik[3], Donald K. Perovich[4], Ian Raphael[4], Mats A. Granskog[3], Kirstin Schulz[5], Matthew D. Shupe[6,7], and Melinda Webster[8]

[1]Woods Hole Oceanographic Institution, Woods Hole, Massachusetts, USA
[2]Institute of Oceanography, Universität Hamburg, Hamburg, Germany
[3]Norwegian Polar Institute, Fram Centre, Tromsø, Norway
[4]Thayer School of Engineering, Dartmouth College, Hanover, NH, USA
[5]Oden Institute for Computational Engineering and Sciences, The University of Texas at Austin, Austin, TX, USA
[6]CIRES, University of Colorado Boulder, Boulder, CO, USA
[7]NOAA Physical Sciences Laboratory, Boulder, CO, USA
[8]Polar Science Center, Applied Physics Laboratory, University of Washington, Seattle, WA, USA
Correspondence to: Madison M. Smith (madisonmsmith@whoi.edu)

**Abstract.** The melt of snow and sea ice during the Arctic summer is a significant source of relatively fresh meltwater. The fate of this freshwater—whether in surface melt ponds, or thin layers underneath the ice and in leads—impacts atmosphere-ice-ocean interactions and their subsequent coupled evolution. Here, we combine analyses of datasets from the Multidisciplinary drifting Observatory for the Study of Arctic Climate (MOSAiC) expedition (June–July, 2020) for a process study on the formation and fate of sea ice freshwater on ice floes in the Central Arctic. Our freshwater budget analyses suggest that a relatively high fraction (58 %) is derived from surface melt. Additionally, the contribution from stored precipitation (snow melt) outweighs by five times the input from in situ summer precipitation (rain). The magnitude and rate of local meltwater production are remarkably similar to those observed on the prior Surface Heat Budget of the Arctic Ocean (SHEBA) campaign, where the cumulative summer freshwater production totaled around a meter during both. A relatively small fraction (10 %) of freshwater from melt remains in ponds, which is higher on more deformed second-year ice compared to first-year ice later in the summer. Most meltwater drains laterally and vertically, with vertical drainage enabling storage of freshwater internally in the ice by freshening brine channels. In the upper ocean, freshwater can accumulate in transient meltwater layers on the order of 0.1 m to 1 m thick in leads and under the ice. The presence of such layers substantially impacts the coupled system by reducing bottom melt and allowing false bottom growth, reducing heat, nutrient and gas exchange, and influencing ecosystem productivity. Regardless, the majority fraction of freshwater from melt is inferred to be ultimately incorporated into the upper ocean (75 %) or stored internally in the ice (14 %). Terms such as the annual sea ice freshwater production and meltwater storage in ponds could be used in future work as diagnostics for global climate and process models. For example, the range of CESM2 climate model values roughly encapsulate the observed total freshwater production, while storage in melt ponds is underestimated by about 50 %, suggesting pond drainage terms as a key process for investigation.

## 1  Introduction

During the Arctic summer, sea ice and snow melt provides a substantial source of relatively fresh water. Precipitation provides an additional source of fresh water during the melt season that is typically small. Meltwater can accumulate in surface or subnivean melt ponds, in brine channels within the ice, or drain to the ocean, where it may accumulate in layers under the ice or in leads, separated from the ocean below by a sharp halocline (e.g., Smith et al., 2022b; Salganik et al., 2023a; Smith et al., 2023), or be directly incorporated into the upper ocean. The magnitude and fate of freshwater associated with sea ice melt are important for the surface energy budget, ice mass balance, ocean structure, and primary productivity, as described below.

Examination of freshwater budgets from prior observational efforts suggested that most freshwater is generated by sea ice surface and bottom melt (Perovich et al., 2021), driven primarily by atmospheric and oceanic heat, respectively. Observations over the past few decades have suggested a shift in partitioning between surface and bottom melt, with bottom melt increasing more rapidly (Perovich and Richter-Menge, 2015). Partitioning varies regionally, with slightly more bottom melt than surface melt observed across most of the Arctic basin. The representation of sea ice melt in global climate models similarly suggests that bottom melt is historically a larger budget term than surface melt (Keen et al., 2021).

Melt ponds form during the summer melt season from the pooling of snow and surface sea ice melt. Their temporal evolution and morphology are controlled by surface topography (Fetterer and Untersteiner, 1998; Petrich et al., 2012; Polashenski et al., 2012; Webster et al., 2015). The seasonal evolution of melt ponds has been a focus of study in recent years due to their significant role in the summer heat budget, primarily through reducing the surface albedo and increasing the transmittance (e.g., Perovich et al., 2002; Light et al., 2022). Observed differences in the timing and extent of melt ponds on multi-year ice compared to first-year ice (Webster et al., 2015; Polashenski et al., 2012; Buckley et al., 2020) suggest variability across ice types in the meltwater budget.

Thin meltwater layers under sea ice and in leads can form in calm conditions during the Arctic summer. The relatively fresh, warm water forms a stable layer on top of the saline ocean, separated by a sharp halocline. A recent review paper (Smith et al., 2023) summarizes observations of these layers, and suggests that they are spatially and temporally heterogeneous but relatively common and persistent across many regions of the Arctic. In fact, these layers were noted as early as the Fram expedition, and have been consistently observed since then (Nansen, 1902; Langleben, 1966; Ehn et al., 2011). Bottom melt rates may be reduced by the presence of strongly-stratified fresher layers under the ice, which limit the transfer of heat from solar radiation (Skyllingstad et al., 2003; Hudson et al., 2013). The presence of under-ice meltwater can further support new sea ice growth during the melt season at the interface between the cold, saline ocean and fresher meltwater layer. These thin layers of ice formed at the interface are commonly called false bottoms (e.g., Eicken, 1994; Notz et al., 2003; Smith et al., 2022b; Salganik et al., 2023a).

The fate of freshwater in the sea ice system ultimately has broad impacts on the physics of the ocean and ice, ecosystems, and biogeochemistry (Smith et al., 2023). Our focus here is on the freshwater input from sea ice and snow melt, which is just a small part the Arctic basin freshwater budget that includes any water source that is less saline than a reference seawater. On the basin scale, freshwater sources may include Pacific and Atlantic water inflows, precipitation, river runoff, ice sheet and glacier

discharge, and sea ice melt, which can be redistributed between Arctic basins and also through sea ice growth, evaporation,
and liquid and solid transport through ocean gateways (Lique et al., 2016; Solomon et al., 2021). Around 90 % of the sea
ice export from the Arctic Ocean takes place in Fram Strait, and the exported ice is an important source of freshwater to the
North Atlantic (Haine et al., 2015). Once in the ocean, meltwater can strongly impact the thermohaline stratification and ocean
circulation (Sévellec et al., 2017). Integration of meltwater into the upper ocean enhances stratification by reducing the density
compared to underlying layers, while discrete near-surface meltwater layers create sharp density interfaces. Stratification due
to sea ice melt and other freshwater sources plays a substantial role on the circulation and ventilation of the Arctic Ocean
(Aagaard and Carmack, 1989). The strength of upper ocean stratification impacts the vertical transport of heat and nutrients in
the ocean (Schulz et al., 2024, in press), accumulation of solar heat (Hudson et al., 2013; Granskog et al., 2015), communities
and productivity (Ardyna and Arrigo, 2020; Gradinger et al., 2010), and can create additional biological stresses (Chamberlain,
2023). The presence of a highly-stratified meltwater layer, in particular, impacts gas exchange between the atmosphere and the
ocean (e.g. Miller et al., 2019).

Recently, Perovich et al. (2021) synthesized data from the 1997–1998 SHEBA experiment to compute a meltwater budget,
addressing the questions: how much meltwater is produced, and what are the relative contributions from different sources over
time? This study computed a budget for multi-year ice from the sea ice perspective, where the fate of the meltwater within
the ocean was not considered. This work raised the question: how will sea ice and upper ocean freshwater budgets change on
and beneath different ice types in a changing Arctic Ocean? Here, we apply a similar approach to quantify a freshwater budget
for the MOSAiC expedition, which took place in summer 2020, observing a mix of first-year ice (FYI) and second-year ice
(SYI). We examine both sources and sinks of meltwater. In contrast to the SHEBA experiment, meltwater sources are adjusted
for ridge contributions using high-resolution observations of ridge fraction and melt, given their prevalence and more rapid
melt than that of level ice (Salganik et al., 2023c), and we additionally estimate internal and oceanic sinks of relatively fresh
meltwater.

## 2  Methods

This study combines various observations made during the MOSAiC melt season to quantify the freshwater budget. MOSAiC
was a year-long drift experiment (October 2019 – October 2020) on and around the Research Vessel Polarstern in the Central
Arctic (Shupe et al., 2020; Nicolaus et al., 2022). The expedition was divided into multiple "legs," with some discontinuity
between for logistical reasons; here we will focus on the melt season observations (Leg 4), which covered late June to the end
of July. During that time, the ice floe adjacent to Polarstern, called the second Central Observatory (CO2), drifted from 82.0° N,
8.3° E to 78.8° N, 2.3° W. The floe was comprised of a mix of FYI and SYI. Autonomous instruments deployed earlier on
the expedition were used to capture the onset of melt prior to the period of study; air temperatures above 0° C and snow melt
onset occurred in late May. The evolution of melt over the observed June-July period roughly progressed onwards with the first
bottom melt at SYI observed on 6 June, the first under-ice meltwater layers at SYI on 16 June, first melt ponds at SYI on 22
June, first under-ice meltwater layers at FYI on 9 July, a large melt pond drainage event on 11-13 July, and ice floe break-up

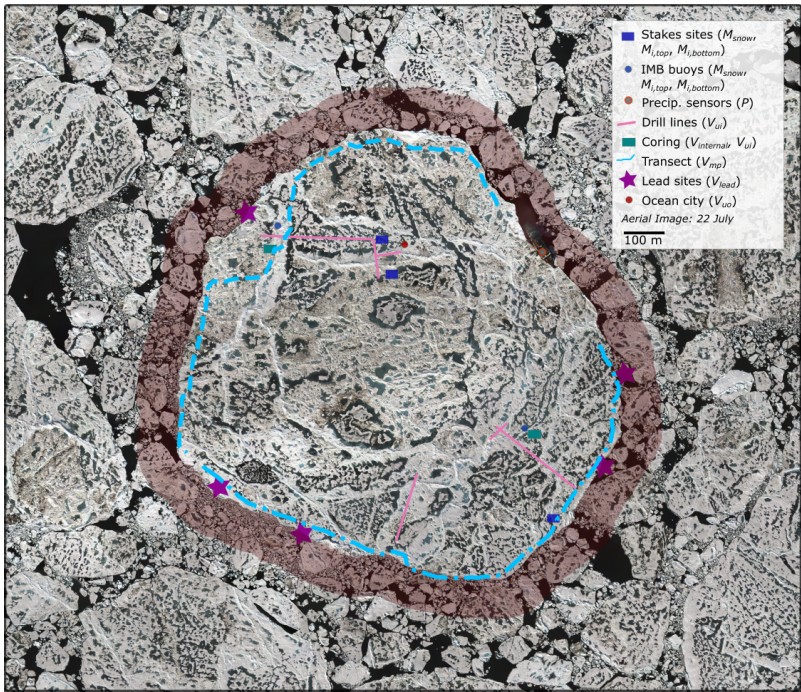

**Figure 1.** Map of locations of measurements used in budgets. The red shaded ring denotes the 100-m border over which the open water (OW) fraction is calculated. Full mass balance transect is shown as blue line; dashed portions denote SYI while dash-dot portions denote FYI. Note that only the locations of ice mass balance (IMB) buoys on the CO2 floe are shown. See Nicolaus et al. (2022) for full map of floe drift.

on 29 July. We use measurements from atmosphere, sea ice, and ocean observatories deployed across the floe to (1) quantify freshwater contributions from sea ice melt, snow melt, and precipitation, and (2) determine the distribution of that freshwater on the sea ice and in the ocean.

## 2.1 Sea ice and upper ocean freshwater budget

Here we define a freshwater budget, where freshwater is defined in terms of the freshwater equivalent relative to a characteristic local ocean salinity. Notably, most of the sources and sinks are not fully fresh, and converting melt to a freshwater equivalent requires some knowledge of the sea ice salinity and density or solid fraction. During MOSAiC, the pre-melt bulk salinity of FYI and SYI was in the range of 2–5 g kg$^{-1}$ (where FYI is typically saltier than SYI) (e.g. Angelopoulos et al., 2022; Salganik et al., 2023a). In some studies, a meltwater volume is alternatively used (e.g., Smith et al., 2022b) but is less appropriate for a budget where salinities of sources and sinks vary. For most terms, the meltwater volume can be converted to an equivalent freshwater volume by a factor of $1 - S/S_{ref}$, where the ratio relates the salinity of the meltwater $S$ to that of the reference seawater $S_{ref}$, taken as 34 g kg$^{-1}$ here. The upper ocean salinity was observed to evolve over time to closer to 32 g kg$^{-1}$ (Fig. B1), while a reference salinity of 35.0 g kg$^{-1}$ is used for freshwater transport monitoring by the Norwegian Polar Institute (2022) and 34.9 g kg$^{-1}$ was used by Aagaard and Carmack (1989), which would result in less than a couple percent error in

freshwater conversion for most of the relevant salinities used here. The ice salinity is taken as a fixed 3 g kg$^{-1}$, which falls within the observational range for both ice types and is between the values of 2 ppt determined to be characteristic for Arctic summer sea ice in Vancoppenolle et al. (2009a) and the assumed value of 4 ppt in Aagaard and Carmack (1989). Fixed densities of 917 and 330 kg m$^{-3}$ are used for sea ice and snow, respectively, following widely used parameterizations and in general
agreement with what was observed during the campaign (Alexandrov et al., 2010; Salganik et al., 2024).

During the melt season, possible sources of freshwater include: snow melt ($M_{snow}$), surface sea ice melt ($M_{i,surf}$), bottom sea ice melt ($M_{i,bottom}$), lateral sea ice melt ($M_{i,lat}$), precipitation (rain; $P$), and condensation. Condensation and evaporation are both vanishingly small as estimated by effectively no net change in the cumulative latent heat flux measured at MOSAiC during the period of study (Cox et al., 2023). The snow, surface sea ice melt, and precipitation over the ice all provide freshwater
sources at the surface of the ice, while bottom and lateral ice melt and precipitation in leads provide freshwater directly to the ocean.

Sinks of meltwater include: storage in melt ponds ($V_{mp}$), under-ice meltwater layers ($V_{ui}$), lead meltwater layers ($V_{lead}$), entrainment into the upper ocean ($V_{uo}$), and internal storage in the ice and ridges ($V_{internal}$). Under-ice meltwater layers can be assumed to generally be a result of vertical drainage, while lead meltwater layers are presumed to be primarily a result of
115 lateral or horizontal drainage. Freshwater is also likely stored in the ice during the melt season through the melt and drainage process by replacement of brine, and re-freezing in ridge keels ($V_{internal}$).

The budget is then defined over the sea ice and upper ocean:

$$M_{snow} + M_{i,surf} + M_{i,bottom} + M_{i,lat} + P = V_{mp} + V_{ui} + V_{lead} + V_{uo} + V_{internal} \tag{1}$$

where calculation of terms are described in the subsections that follow. Here, the budget terms represent a cumulative volume
relevant to the beginning of the melt season (or, the start of available data). Many source terms are initially calculated as a rate which is then summed to calculate a cumulative value, while many sink terms are calculated directly as a volume. We assume a one dimensional budget, where volumes are scaled by the area (m$^3$/m$^2$), simplifying to a unit of length (m). The area the budget is calculated over is generally the MOSAiC CO2 floe (Figure 1), plus an additional 100 m border to include the fraction deposited into adjacent leads (Fig. A1). Thus, all terms are scaled by the open water or ice-covered fraction. This represents a
key difference from the budget calculated in Perovich et al. (2021), which calculated a 1D budget of ice-covered area only.

The open water fraction is determined by thresholding aerial orthomosaics of the floe (Fuchs, 2023; Neckel et al., 2023). The estimate is made over the area of the floe plus a 100-m ring (red shaded areas in Fig. A1). Images to make such estimates are only available for three dates — 30 June, 17 July, 22 July — so a linear interpolation is used for dates between to enable continuous budget calculations. Open water fraction estimates on 30 June, 17 July, and 22 July are 3.3 %, 9.6 %, 5.2 % (Fig.
A1), respectively. Comparisons with satellite product estimates of sea ice concentration from Bootstrap (Comiso, 2000) and NASA Team (DiGirolamo et al., 2022) methods at the MOSAiC location over the same time period show that these values are within the range of observational estimates, which have a standard uncertainty of around 20 % for summertime retrievals, corresponding to a couple percent error for the airborne open water fraction estimates.

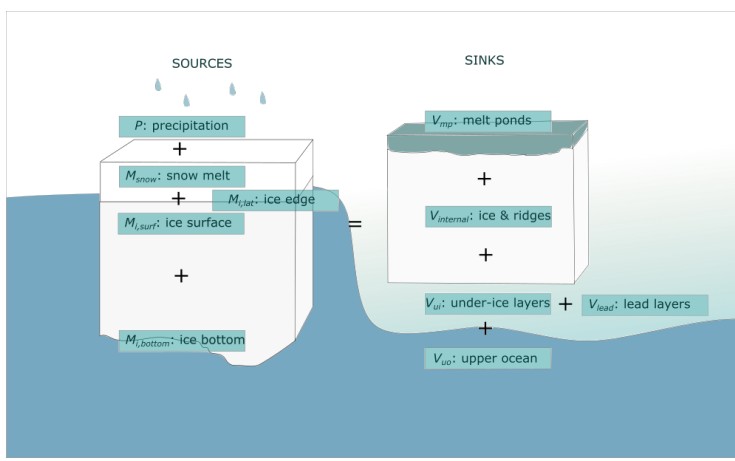

**Figure 2.** Schematic of key terms of the freshwater budget, with sources on left and sinks on right. Ice bottom source term includes an adjustment for the ridge fractional coverage. The dark blue background represents the shifting depth of the interface of saltwater with fresher water.

The floe (Fig. 1) contains a combination of FYI and SYI (Guo et al., 2023; Kortum et al., 2024). We assess differences in budget terms over known FYI and SYI portions of the floe, where possible, including the sources from melt and the melt pond sink term ($V_{mp}(t)$). However, it is not possible to complete a full budget for both due to insufficient data to address all terms for the separate ice types. Additionally, as the upper ocean interacts with both FYI and SYI, attempting to separate the budget of these ice types is not representative of the reality of such a composite floe.

## 2.2 Freshwater sources

### 2.2.1 Snow and ice melt $M_{snow}$, $M_{i,surf}$, $M_{i,bottom}$

Melt of snow and sea ice are estimated from a combination of two methods which use the approach of measuring the evolution of interfaces: sea ice mass balance stakes (hereafter, stakes) and ice mass balance buoys (IMBs). Estimates of snow and sea ice melt terms from the two methods are compared, and the values are taken as an average, as shown in Figure 3.

During the observational period, arrays of stakes were deployed on three areas of the floe to provide a representative sample (Raphael et al., 2022). For example, 38 % of the stakes were in melt ponds at some point with a peak melt pond fraction of 23 %, which generally agrees with the satellite estimates from Niehaus et al. (2023) and Webster et al. (2022). Terms are averaged across all stakes to provide estimates from 26 June onwards. More details can be found in Raphael et al. (2024). As the installation of stakes in late June missed initial melt onset, the melt of snow and ice prior to 26 June was estimated using an IMB buoy at a site known as L2, 0.7 degrees east of the floe on 1 July, which had representative ice and snow thickness values (Perovich et al., 2023; Raphael et al., 2024). The SIMB3 buoy suggested that all change in surface elevation prior to 26 June was due to snow melt. A comparison of meltwater production from snow and ice melt between L2 IMB and stakes from

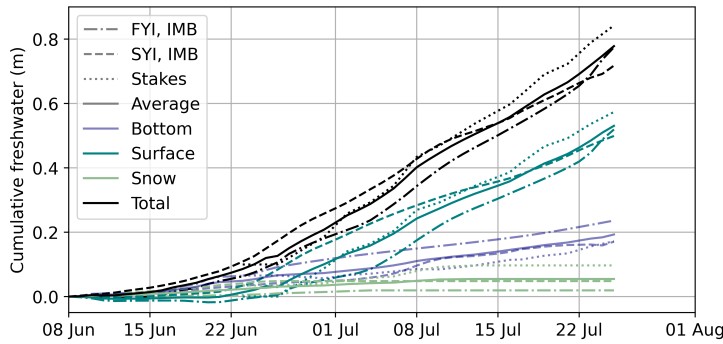

**Figure 3.** Estimation of sea ice and snow melt freshwater source terms from stakes (dotted lines) and IMB buoys. IMB buoys are separated into FYI (dot-dashed lines) and SYI (dashed lines). The average across all methods (solid lines) is used as the best estimate of each term including bottom melt (navy), surface melt (teal), and snow melt (green), in subsequent analysis.

26 June – 29 July (not shown) are quite consistent, suggesting that this provides a robust estimate of meltwater production for level ice, and that the L2 observations between melt onset and 26 June are likely representative of the CO2 floe.

A total of 18 IMB buoys were operational during Leg 4 of MOSAiC on the CO2 and surrounding floes (Rabe et al., 2024). Of these, 12 were deployed in SYI, 3 in FYI, and 3 undefined or other, with maximum ice thicknesses ranging 1.55–5.94 m, and maximum snow thicknesses ranging 0.13–0.62 m (with the largest for either being an outlier from a buoy in a first-year ridge). For the purpose of snow and ice melt calculations, the dataset was restricted to IMBs that are on a known ice type that covered the entire period through 25 July. This left 5 IMB buoys in SYI, with maximum ice thicknesses 1.71–2.92 m and maximum snow thicknesses 0.15–0.36 m and 3 IMB buoys in FYI with maximum ice thicknesses 1.55–1.86 m and maximum snow thicknesses 0.14–0.20 m. Each set includes 1 buoy in the MOSAiC CO2.

Recent work has shown that the melt of ridge keels provides a larger and more rapid input of meltwater compared to level ice (Salganik et al., 2023c). Most mass balance measurements used in these estimates were made over relatively level ice (avoiding ridge keels). Further, since melt of ridge keels is highly spatially heterogeneous it is not well captured by point measurements from stakes or IMBs. We make an adjustment to the bottom ice melt estimate based on the ridge (keel) fraction, $A_{ridge}$, and the ratio of ridge keel melt to level ice melt rate, $R_{ridge/level}$

$$M_{bottom} = (M_{bottom,level} \cdot (1 - A_{ridge})) + (R_{ridge/level} \cdot M_{bottom,level} \cdot A_{ridge}) \tag{2}$$

where $R_{ridge/level}$ is assumed as a constant 3.8 based on results from Salganik et al. (2023c) using repeated multibeam sonar surveys. $A_{ridge}$ is estimated by correcting the ridge sail fraction with a sail to keel width ratio. The sail fraction is estimated as 8 % using the using Airborne Laser Scanning data from July with a 0.6 m freeboard threshold (ALS; Hutter et al., 2021). The sail to keel ratio was estimated as 2.7 using co-located ALS and remotely operated underwater vehicle sonar data with the same freeboard threshold (Hutter et al., 2021; Salganik et al., 2023c). This ratio is slightly lower than previously estimated ratios of Arctic FYI ridges of 3–7 (Strub-Klein and Sudom, 2012) and 4.1 (Guzenko et al., 2023). This suggests a keel areal

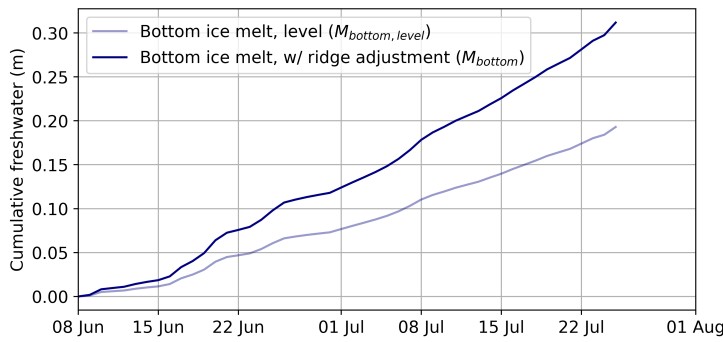

**Figure 4.** Impact of ridge keel coverage adjustment on calculation of cumulative bottom ice melt freshwater production. The initial calculation using mass balance buoy and stakes only, from relatively level ice ($M_{bottom,level}$), is shown in lighter transparent blue, and the total estimate of bottom ice meltwater with the keel melt adjustment ($M_{bottom}$) is in navy.

fraction $A_{ridge}$ of 22±4 %. This is in agreement with pan-Arctic estimates of 12–20 %, assuming a typical keel width of 36 m (see Strub-Klein and Sudom, 2012) and average ridge sail spacing of 250–300 m (e.g., Mchedlishvili et al., 2023; Zhang et al., 2024). The impact of this adjustment is shown in Figure 4.

### 2.2.2 Lateral ice melt, $M_{i,lat}$

As direct measurements of lateral melt were not made during MOSAiC, we provide an estimate of the approximate contribution from lateral melt using the change in floe size estimated from orthomosaics (Neckel et al., 2023). Estimates of the floe size were available for three dates: 30 June, 17 July, and 22 July. Approximate lateral melt contribution between these dates was estimated by multiplying the change in floe area by the average ice thickness from mass balance surveys (Itkin et al., 2023), in order to get the change in floe volume, and scaling by the initial area. We note that change in floe size may be a result of both thermodynamic loss (lateral melt) as well as dynamic loss (breaking of the floe); here we assume that all loss in area is due to thermodynamic processes. The error in this rate estimate is likely large, but it contributes a small fraction to the total freshwater estimate.

### 2.2.3 Precipitation $P$

The liquid-phase precipitation (rain) estimates were made in two ways. First, an optical measurement of precipitation was made by a Present Weather Detector (PWD) installed above the bridge of the ship (Kyrouac and Holdridge, 2019). This sensor has been shown to be a superior measure of snowfall in the winter compared to other precipitation gauges at MOSAiC as it minimized the impacts of blowing snow (Matrosov et al., 2022), although it is not clear how this result translates to liquid-phase precipitation in summer. Additionally, since it was installed onboard the vessel, it was operated nearly continuously providing an uninterrupted dataset. The second approach was to identify periods of liquid-phase precipitation using a multi-sensor approach based on cloud radar and other instruments operated onboard Polarstern (e.g., Shupe, 2007, 2022), and then to apply an

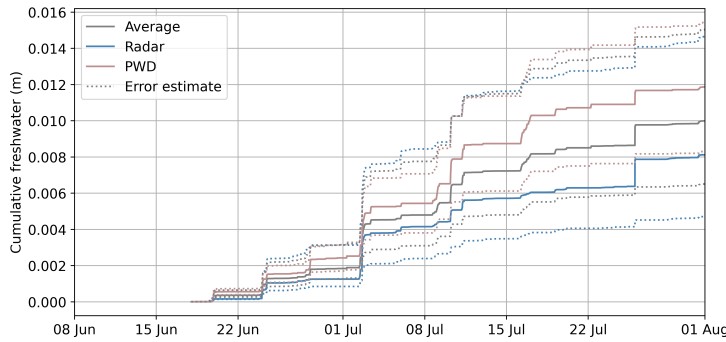

**Figure 5.** Cumulative freshwater precipitation in meters from late June to end of July, 2020. Estimates from radar (pale blue) and PWD (pale red) are averaged together (grey) to give a best estimate of precipitation for use as $P$. The error estimates given minimum and maximum values (dotted lines) from the two methods are averaged together to give the best estimate of total observational uncertainty.

appropriate radar reflectivity based power-law retrieval (e.g., Chandra et al., 2015). The radar reflectivity measurements came from a Ka-band ARM Zenith Radar (KAZR) operated from Polarstern, with precipitation retrievals being applied to measure-ments at a height of 190 m above the surface, which is the lowest range gate from the radar data that avoids any near-surface influences. Precipitation is assumed to be consistent below that height.

Here, we use an average of the two methods (solid grey line in Fig. 5) to quantify the rate and cumulative accumulation of in situ precipitation. The uncertainties are quantified as the average of minimum and maximum uncertainty values from the radar and PWD methods. Uncertainties in the radar-based approach may come from three primary sources. First, the calibration of the radar reflectivity is assumed to contribute uncertainty up to 3 dBZ. The second source of uncertainty is the inversion of radar measurements to retrieve a precipitation rate, where we include here the highest and lowest values retrieved from approaches in the literature. The third source of uncertainty is associated with potential mis-identification of conditions (rain vs. drizzle), which is ignored here as most conditions were representative of rain. The upper bound on radar-based precipitation is thus estimated as the retrieval method with the highest estimated accumulation with a reflectivity that was measured plus 3 dB. The lower bound on radar-based precipitation is estimated as the retrieval method with the lowest estimated accumulation using a reflectivity that was 3 dB less than what was measured (where both are shown as dotted blue lines). The uncertainties associated with the PWD have been less well quantified. Based on comparisons given in the literature we approximate the uncertainty as 30 %, so a $\pm$ 30 % uncertainty is applied to the observations (red dotted lines). PWD gives a higher overall estimate than the radar method, but the estimated average 10 mm rain accumulation falls within the plausible range from both methods.

## 2.3 Freshwater sinks

### 2.3.1 Melt ponds $V_{mp}$

Melt pond depths and fraction are estimated using the modified Magnaprobe estimates along the Leg 4 transect (Itkin et al., 2021; Webster et al., 2022), excluding any special surveys or albedo lines. Estimates include both the depth of standard melt

ponds and those defined as subnivean ponds. We include results for the separation of transect data into SYI and FYI, where delineation of the floe into ice types was determined by observers on the floe (SYI is denoted by dotted blue line in Fig. 1; FYI is denoted by dashed line; the remaining solid fraction of the line is an unclassified ice type but is included in the full transect calculations; Webster et al., 2022). Conversion to freshwater equivalent assumes that melt ponds have a practical salinity of 1.1 as an average of measurements made in July (Lange et al., 2022), which was generally consistent with a full observed range of 0.5–2 across ponds and ice types (Oppelt and Linhardt, 2023).

### 2.3.2 Meltwater layers under ice $V_{ui}$

Freshwater in under-ice meltwater layers is estimated primarily using layer thicknesses from all of the 18 IMB buoys operable during the melt season (see description of buoys in Section 2.2.1). While the IMB buoys captured a range of sea ice types, we note that these may constitute an unrepresentative sample. The top and bottom extent of the meltwater layers, when present, are determined based on location of the ice-meltwater interface and meltwater-ocean interface using the rate of change after heating cycles and ambient temperature, respectively (Jackson et al., 2013). As the thermistor strings on the buoys have 2 cm spacing in sensors, an accuracy of 2 cm can be assumed. IMB buoys provide estimates of temperature and thus interfaces every 6 hours, which we average to calculate a daily value. We calculate the layer thickness as an average across all instruments where a layer was observed on that day.

Estimates of under-ice meltwater layer thickness from IMB buoys are compared in Figure 6 with those measured along drill lines (orange lines in Fig. 1) by deployment of a YSI Professional Plus probe measuring temperature and conductivity (Smith et al., 2022b), and those made at the coring site (teal box in Fig. 1, Salganik et al., 2023a). The coring site estimates are consistent with, but smoother than, the estimates from IMB buoys, while the YSI estimates seem to provide a "lower-bound" from a different region of the floe (SYI). We note that a layer of under-ice meltwater was also observed at the Ocean City hole using a microstructure profiler from 28 June to 13 July (Smith et al., 2023; Schulz et al., 2022), but those observations are not included here due to suspected biases associated with possibly artificially accelerated meltwater drainage and the location in SYI.

The conversion from meltwater layer thickness (Fig. 6) to equivalent freshwater thickness accounts for the salinity of the layer, which is not fully fresh, and the spatial coverage under the ice. Here we assume a constant salinity of 8 based on an approximate temporal best fit of all salinity estimates included in Smith et al. (2022b). A constant fractional coverage of 21 % is assumed (Salganik et al., 2023a) as information is too sparse to justify more temporal variability in coverage, and this value is approximately consistent with the estimates in Smith et al. (2022b). This is lower than the fractional estimate based on the fraction of IMBs in which meltwater layer was observed (8 out of 18; 44 %), which may be biased high due to the fact that IMB installation typically occurs in areas of undeformed ice far from floe edges, which are more prone to under-ice meltwater layer formation.

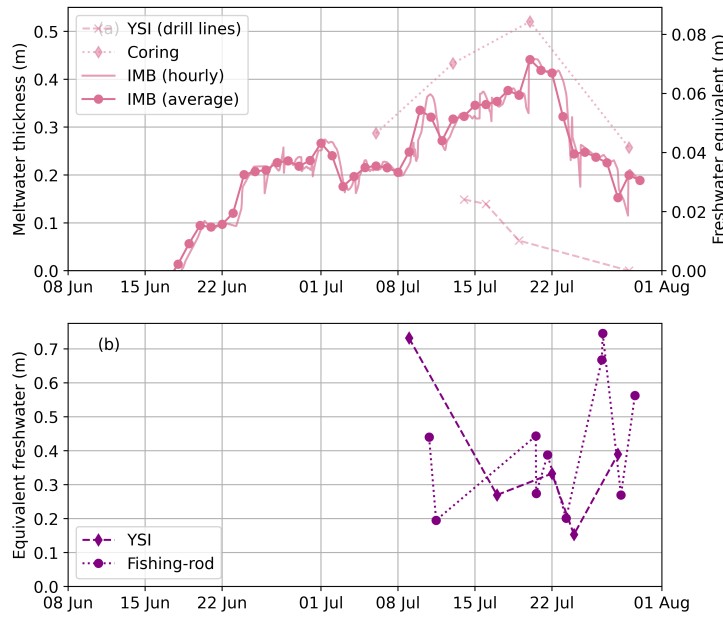

**Figure 6.** (a) Comparison of estimates of under-ice meltwater layer thickness (left axis) and equivalent freshwater thickness (right axis; $V_{ui}$) from different observational methods, prior to scaling for coverage. Estimates from drill lines (dashed) and FYI coring site (dotted) are shown for comparison to the average layer thickness from IMBs (solid line). (b) Estimates of freshwater layer thickness in leads $V_{lead}$ (prior to scaling for open water fractional coverage) from YSI (dashed line, diamonds) and fishing-rod CTD (dotted line, circles).

### 2.3.3 Meltwater layers in leads $V_{lead}$

Freshwater equivalent estimates for lead meltwater layers are approximated as the average of all measurements available from profiling by a YSI Professional Plus probe (as used for under-ice measurements; Smith et al., 2021a) or fishing-rod CTD for a given day (Karam et al., 2023) (Figure 6). Here, profiles were generally made from the lead edge or a kayak, to minimize disruption of the surface stratification. Calculation of the equivalent freshwater layer thickness results in estimates that range from 72 %–150 % of the depth of the maximum density change, and equivalent freshwater estimates are more consistent between the two instruments than estimates of meltwater layer thickness (YSI and fishing-rod CTD). Lead meltwater layer estimates are scaled by the open water fraction (Figure A1), whereas all sea ice-based estimates are scaled by the ice-covered fraction.

### 2.3.4 Upper Ocean $V_{uo}$

While sub-daily estimates of upper ocean temperature and salinity are available for the duration of the period (Schulz et al., 2024, in press), estimates of freshwater input to the upper ocean are complicated by the drifting nature of the experiment. Notably, in this period the MOSAiC floe drifted into a regime with lower ambient seawater salinity and higher river water

fractions (see Schulz et al., 2024, in press) (Appendix B), which dramatically increased the apparent "freshwater content" in a manner that is not entirely related to local inputs from sea ice. As a result, we do not directly estimate entrainment of freshwater into the upper ocean, and rather treat this term as a residual.

### 2.3.5 Internal storage $V_{internal}$

Sea ice emits salt to the upper ocean both during growth and melt, directly impacting the salt content of the surrounding seawater through desalination processes. These include gravity drainage and flushing during the melting season. Gravity drainage is the exchange of the dense brine in the ice with seawater by convective overturning triggered by a heat-induced increase in ice permeability. Flushing is the replacement of the salty brine in the ice with surface meltwater percolating through the ice (Notz and Worster, 2009). In relative terms, both processes reduce the freshwater equivalent by releasing a salt mass $\Delta m_{salt}$ from the ice into the ocean and form thus a sink term in the freshwater budget that we consider with $V_{internal}$. Following the scheme to start from the surface meltwater, $V_{internal}$ is derived here from the volume difference of the freshwater equivalent of the surface meltwater ($V_{MW,surf} = M_{snow} + M_{i,surf} - V_{mp}$) before draining through the ice ($V_{MW,equ}(in)$) and after ($V_{MW,equ}(out)$):

$$V_{internal} = V_{MW,equ}(in) - V_{MW,equ}(out) \tag{3}$$

The freshwater equivalent of the surface meltwater before draining into the ice was obtained by scaling the available surface meltwater $V_{MW,surf}$ using:

$$V_{MW,equ}(in) = V_{MW,surf} \cdot \left(1 - \frac{1.1\text{g kg}^{-1}}{S_{ref}}\right), \tag{4}$$

for which we assume a constant salinity of 1.1 g kg$^{-1}$ of the surface meltwater (Lange et al., 2022) and the previously defined reference salinity $S_{ref} = 34$g kg$^{-1}$.

After draining through the ice, the volume of meltwater remains constant, i.e. isochoric, but the salt content changes, and thus the freshwater equivalent changes to:

$$V_{MW,equ}(out) = V_{MW,surf} \cdot \left(1 - \frac{S_{MW}}{S_{ref}}\right) \tag{5}$$

with the increased salinity of the drained meltwater:

$$S_{MV} = \frac{\Delta m_{salt}}{(\Delta m_{salt} \cdot 1000 + m_{MW,surf})} \tag{6}$$

The generalization across all desalination processes was made to be able to derive a time series of $V_{internal}$ both with measurement data, combining the observed total loss of salt in ice cores $\Delta m_{salt}$ with surface meltwater, and with simulations, which fully resolve all individual desalination processes using the multiphase thermodynamic single-column model Semi-Adaptive Multi-phase Sea-Ice Model (SAMSIM; Griewank and Notz, 2013). The model was initialized and run separately with ice core data from FYI and SYI from early May (Oggier et al., 2023a, b) and then forced with a set of boundary conditions. Boundary conditions used a merged time series of observed atmospheric boundary fluxes (Pirazzini et al., 2022), ERA-5 data

as gap fillers (Hersbach et al., 2017), a rather small constant oceanic heat flux of 1 W m$^{-2}$ and an initialized snow thickness of about 22 cm based on observations. Changes in salinity of the underlying ocean water due to the formation of the meltwater layer are not considered in the model. Given these forcings, the model simulated ice evolution similar to the observed ice core measurements. Bulk brine volume estimates from IMB buoy temperature observations are very similar to those from coring observations in magnitude and temporal evolution on both FYI and SYI (Fig. C2), giving confidence to the applied estimates.

Meltwater can also refreeze in ridge keels during summer (e.g., Salganik et al., 2023b), which may provide a transient sink in opposition to the source from accelerated ridge keel melt. We do not make explicit estimates of it here as reliable pre-melt season salinities are not available. However, we expect the term to be relatively small as the keel macroporosity in June was already very low (4–6 % for a ridge on the same floe) suggesting low meltwater mass fraction (Salganik et al., 2023b).

## 3 Results

### 3.1 Freshwater sources

Estimates of sea ice and snow source terms from methods and ice types are compared in Figure 3. Initially, there is approximately 2–3 times more surface and snow melt on SYI than FYI, but around 2 times more bottom melt on FYI than on SYI. By the end of the observation period (late July), the cumulative freshwater produced on both ice types approximately converges, with the larger bottom melt of FYI (40 % greater; 6 cm) mostly compensating for the larger snow melt on SYI (2.5 times greater; 3 cm). The estimates from stakes sit mostly between the estimates for FYI and SYI from IMB buoys, though with a somewhat larger freshwater contribution from snow melt and surface melt estimated by the end of the period. In general, the relative ranking of terms is consistent across methods and ice types.

The bottom melt on relatively level ice (blue solid line in Fig. 3) is adjusted for the higher melt rate observed on ridge keels in Figure 4, following Eqn. 2. Ridge keel melt is estimated to contribute cumulatively around 0.12 m of freshwater by the end of the period, increasing cumulative bottom melt estimates by over a third (navy line in Fig. 4).

All source budget terms are summarized in Figure 7. The rate of freshwater input from all terms appears quite episodic, as dictated by atmospheric and ocean forcing. The contribution from surface ice melt is initially close to zero as solar energy is likely focused on snow melt, and some buoys observed re-freezing of melt ponds. The surface and bottom sea ice melt ($M_{i,surf}$, $M_{i.bottom}$) generally accelerate over the course of the observation period. In contrast, the contribution from snow melt ($M_{snow}$) ceases on 13 July, when nearly all snow had melted on level ice (Macfarlane et al., 2023; Webster et al., 2022). Lateral melt estimates ($M_{i,lat}$) suggest that it is a low fractional contribution through the period (less than 5 % of the total). The contribution from precipitation is episodic, with a total cumulative contribution barely over 1 cm. The largest overall contribution to the cumulative freshwater is from surface sea ice melt, while the precipitation provides an almost negligible contribution.

Both precipitation-related source terms ($M_{snow}$ and $P$) in total contribute less than 7 % of freshwater to the cumulative total by the end of the period. Given the uncertainty in both initial snow thickness and precipitation of up to 50 % (Itkin et al., 2023, ; Fig. 5), we estimate this could range from 4–10 %. Figure 8 compares the relative contributions from these terms. The ratio

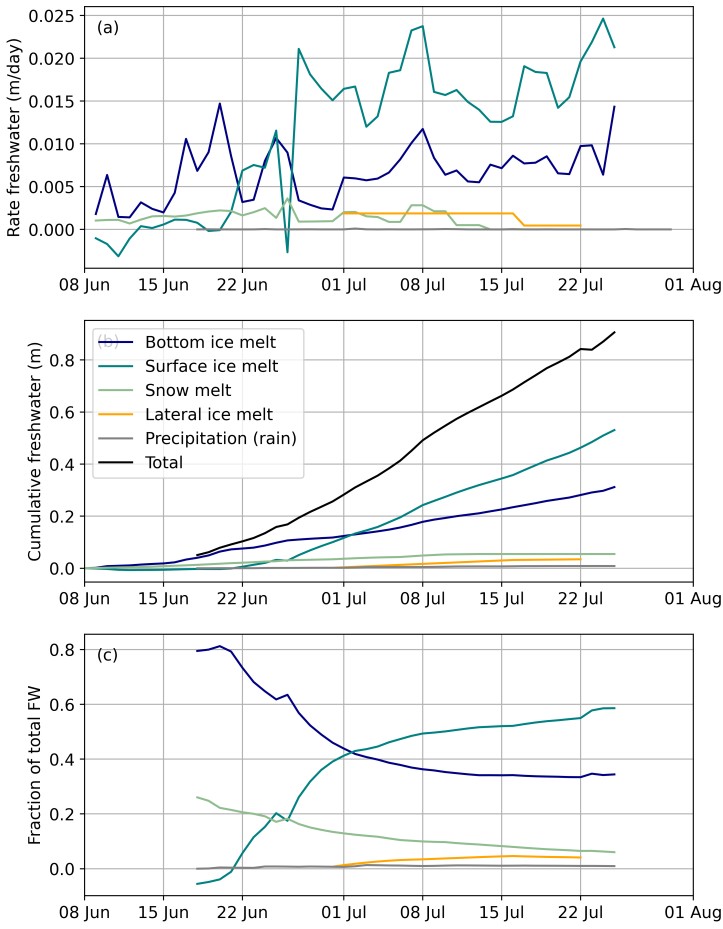

**Figure 7.** Summary of freshwater production (freshwater equivalent) from early-June to late July, 2020. (a) Rate of freshwater production from bottom ice melt (navy), surface ice melt (teal), snow melt (green), lateral melt (yellow), and precipitation (grey). (b) Cumulative freshwater from early June to late July, including the total of all terms (black). (c) The fraction of total freshwater (FW) contributed by each term.

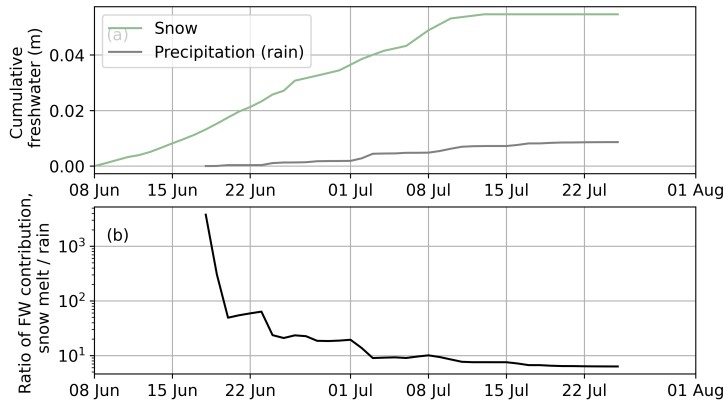

**Figure 8.** Comparison (a) and relative contribution (b) from stored precipitation (snow; green) and in situ precipitation (grey).

is very high initially ($>$100), as snow melt is rapid and little rain has yet been recorded. By the end of the observation period, stored precipitation (snow) contributes around six times as much to the freshwater budget as in situ precipitation (rain). The relative fraction is likely to continue to decrease past the period of this time series, as precipitation may continue in August but the snow has virtually all melted.

## 3.2 Freshwater sinks

Figure 9b shows that storage in melt ponds is notably higher on SYI (dashed line) compared to FYI (dash-dot line). Equivalent freshwater storage is over twice as large on SYI at times. The temporal evolution on FYI is less extreme compared to that on SYI through the middle of the observation period, likely owing to the earlier formation of lateral drainage channels on FYI (Webster et al., 2022). However, the temporal evolution is largely similar between the ice types, suggesting that the drivers of sources (meltwater input) and loss via drainage are not independent. Some of the temporal variability on the FYI portion of the transect from 20 July onward could be biased by necessary relocation of the transect as melt ponds melted completely through (Webster et al., 2022). The substantial decrease 10–14 July on the SYI was associated with a large drainage event that was visible from the floe and in aerial imagery and resulted in a large reduction in both melt pond fraction and depths (Webster et al., 2022) and increase in ice freeboard (e.g., Salganik et al., 2023c). The average melt pond freshwater storage along the full transect (solid line) is used for the comparison of sink terms in subsequent analysis.

We note here that the estimates of freshwater in melt ponds during MOSAiC, as quantified from transect data, may be biased low due to the location at the periphery of the ice floe (Figure 1), where lateral drainage is greater. Specifically, estimates of freshwater equivalent from bathymetric reconstructions of orthomosaics suggest approximately twice as much freshwater stored in melt ponds in the first half of the period (30 June and 7 July) compared to transect observations (Fuchs et al., 2024). Further investigation into these differences suggest that this is likely mainly due to differences in observed areas instead of methodological differences (Fuchs et al., 2024). The orthomosaic estimates are not used here due to sparse temporal coverage, and the reader is referred to Fuchs et al. (2024) for more details on these observations.

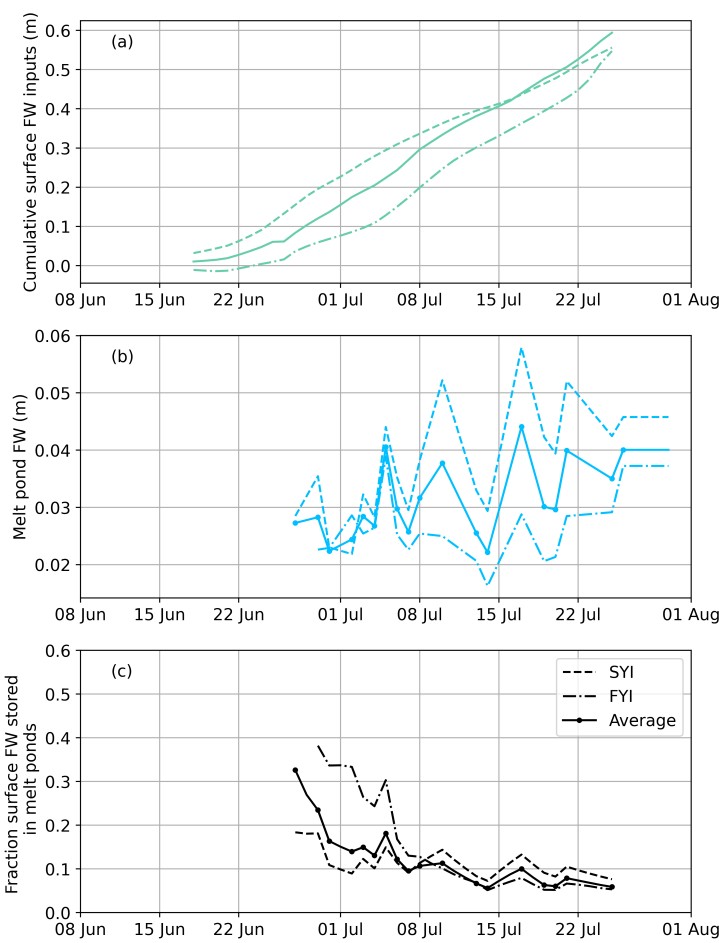

**Figure 9.** (a) Cumulative surface freshwater input on SYI (dashed) and FYI (dash-dot) from combined surface ice melt, snow melt, and precipitation. (b) Melt pond equivalent freshwater thickness for SYI and FYI. Solid line shows the equivalent thickness for the full transect on each date, while the dashed and dash-dot lines show equivalent thicknesses over just the SYI and FYI sections, respectively (see Fig. 1). (c) The ratio of freshwater in melt ponds to the freshwater input (b/a) provides an estimate of the melt pond fractional storage. Following the initial peak in storage on 27 June, the storage (c) gradually decreases throughout July largely due to the steady increase in input from surface meltwater (a) which is not matched by a comparable increase in equivalent pond volume (b).

Internal storage of freshwater due to desalinization based on observational (coring) data and modeling is shown in Figure 10. The fully-resolving model indicates a higher $V_{internal}$ for both ice types over the entire summer melt and indicates two main desalination periods: the first in late May through warming of the ice interior, during a period when there is a gap in coring and other observational datasets, and a second period starting mid-July. The observations additionally capture the rapid increase in FYI storage in mid-July. While both ice types showed the same amount of total surface melt over the observational period (Fig. 3), FYI typically desalinates more and stores more freshwater. We thus use the estimate from FYI cores (dark brown dot-dashed

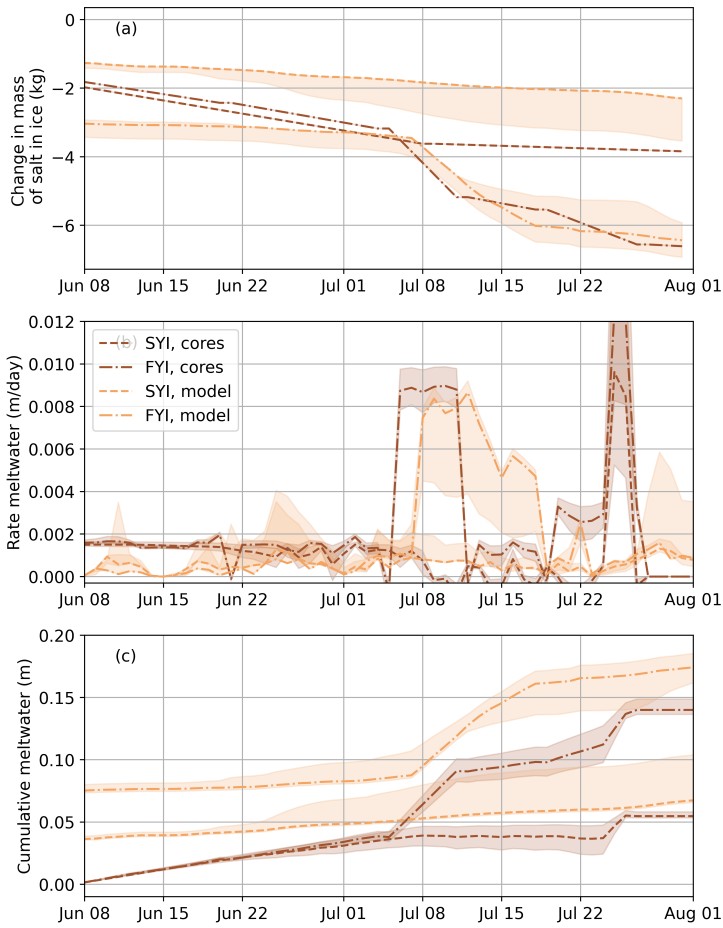

**Figure 10.** Estimates of salt internally in ice and contribution to freshwater sink. (a) The change in mass of salt in ice, relative to the start of the melt season, is used to calculate the (b) rate and (c) cumulative freshwater stored internally in the sea ice ($V_{internal}$). Estimates are shown based on coring observations (dark brown) and 1D model (light brown) for SYI (dashed) and FYI (dot-dashed).

line in Fig. 10b) as the timeseries of internal freshwater storage $V_{internal}$ in subsequent comparisons, providing a likely upper bound on the term. Sensitivity of these terms is approximated using runs where forcing terms are given the following ranges:

ocean heat flux from 0.5–10 W/m$^2$; reference salinity $S_{ref}$ from 32–34.9 g kg$^{-1}$ (Schulz et al., 2024, in press; Norwegian Polar Institute, 2022); meltwater salinity $S_{mw}$ from 0.6–1.9 g kg$^{-1}$ based on the range of observed melt pond salinities (Smith et al., 2022b; Lange et al., 2022). The range of all runs for coring and model calculations are shown as shading in Figure **??**, and suggest errors from around 5–100 %. For the FYI core estimates used in subsequent calculations, this represents a 9 % range. We do not explore in detail the relative sensitivity of the estimated meltwater value to the input terms, but results suggest

the greatest impact is that of the meltwater salinities.

()

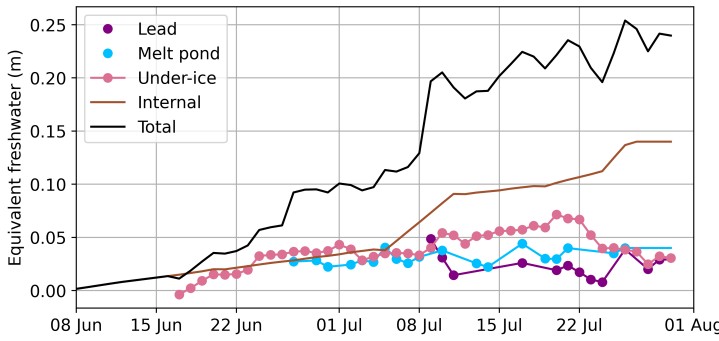

**Figure 11.** Summary of meltwater sinks (freshwater equivalent) from mid-June to late July. Total (black) includes estimated freshwater in leads (purple), melt ponds (cyan), under-ice (pink), and internal in the ice (brown).

Quantified freshwater sink terms both on the ice and in the upper ocean are combined in Figure 11. Melt ponds and under-ice meltwater layers were observed from near the beginning of the record, with lead meltwater layers only first being observed on 8 July. Thus, under-ice layers and melt ponds are the dominant sink in early July. Under-ice meltwater layers constitute a significant fraction of the observed sinks, with melt ponds and leads representing approximately the same volume from middle to end of July. In mid-July, internal storage in the sea ice becomes the dominant sink, with over 0.10 m equivalent freshwater storage by the end of the observation period. The total freshwater equivalent stored in the quantified sinks (black line in Figure 11) peaks at 0.25 m in late July.

### 3.3 Sources vs. sinks

By comparing the melt pond term with the surface sources, we can understand the efficiency of melt ponds as sinks. Sources for melt ponds include snow melt, sea ice melt, and precipitation, which can be determined from the results in Fig. 7 to calculate the cumulative surface freshwater input (Fig. 9a). Dividing this by the melt pond volume, $V_{mp}$, following (Perovich et al., 2021), gives an estimate for the fractional storage of surface freshwater in melt ponds in Figure 7c. During their initial formation, melt ponds are an efficient sink for surface meltwater, with over 45 % estimated on 27 June. This rapidly decreases with a decrease in freshwater in melt ponds in late June, and fractional storage remains below 20 % throughout July.

Figure 9c also shows differences in the relative melt pond storage of ice types. Initially, FYI stores more surface meltwater than SYI, but SYI stores a higher relative fraction after 7 July. As both surface meltwater inputs and melt pond freshwater volume are higher throughout the summer on FYI, this temporal shift largely results from a decline in the rate of surface freshwater production on FYI over time.

Comparing the total quantified freshwater sinks with estimates of sources provides insight into the full freshwater budget and the residual that is deposited into the ocean. By the end of the observation period, 0.68 m or 75 % of the estimated sources (Fig. 7) are unaccounted for by observed sinks (Fig. 11). This suggests that a majority of local freshwater from sea ice and snow melt is ultimately entrained into the upper ocean. We can use this to make some indirect inferences about the residence

time of freshwater within the various sinks. While the magnitude of freshwater within lead, melt pond, and under-ice sinks

does not significantly grow over time, freshwater sources provide relatively consistent new inputs. It is thus likely that most of the freshwater that ends up in the upper ocean passes through one of these other sinks first. In other words, initial increases in the under-ice freshwater sink suggests that the flux of freshwater from sources is outpacing the mixing into the upper ocean, while the decrease later in the observation period suggests that the mixing is outpacing new inputs.

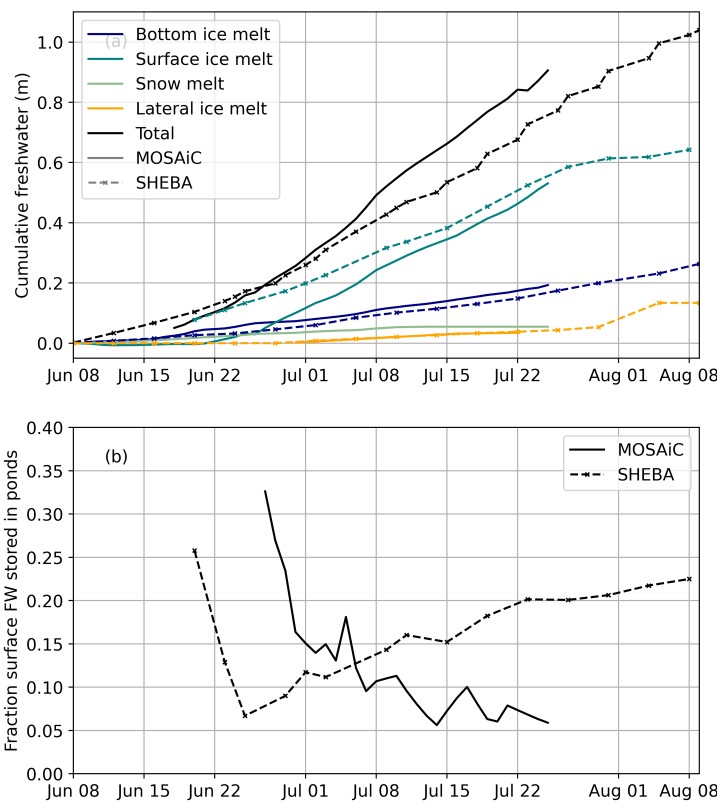

**Figure 12.** Comparison of meltwater budget terms from MOSAiC (solid) and SHEBA (dashed). (a) Cumulative meltwater source terms, noting that snow is not shown for SHEBA, and (b) fractional melt pond storage.

## 4 Discussion

### 4.1 Comparison with prior observations: SHEBA

Perovich et al. (2021) provided one of the first comprehensive budgets of meltwater in Arctic sea ice using observations from the 1997–1998 SHEBA datasets. Notable differences between MOSAiC and SHEBA include that sea ice during MOSAiC was a mix of FYI and SYI, while SHEBA occurred on dominantly MYI. Additionally, MOSAiC was in the Central Arctic, approaching Fram Strait by the time of the melt season, while SHEBA occurred in the Beaufort Sea. Some changes in conditions can also be expected as a result of the 22 years separating the expeditions. More extensive quantification of thin meltwater layers on MOSAiC (e.g., Smith et al., 2022b, 2023) also allowed calculation of sinks.

Figure 12a compares the cumulative freshwater inputs from major sea ice and snow sources. The total cumulative input is remarkably similar between the two by the end of the comparison period, despite the differences in conditions. Notably, the contribution from bottom melt closely tracks between the two sets of observations, despite the significant adjustment for contribution from ridges on MOSAiC. The multiyear ridges during SHEBA were smoother and so produced less meltwater

than typical FYI ridges, with only 1.6 times more bottom melt at ridge keels (Perovich et al., 2003). The surface melt was a larger fraction of melt on both campaigns, but accelerated more rapidly earlier in the melt season during SHEBA. The lateral melt is a minor fraction (∼5 %) on both campaigns by the end of July. In both, the contribution from summer precipitation is small compared to other terms, and can be considered negligible (not shown here).

Fractional storage in ponds drops sharply early in the melt season following initial drainage due to melt pond-ocean connection on both MOSAiC and SHEBA. The storage continues to decrease over time on MOSAiC in contrast to SHEBA, where fractional storage somewhat rebounded following the initial drainage (Figure 12b). The decrease on MOSAiC is in large part due to the rapidly increasing meltwater input, which outpaces the storage in ponds (which also increases; see Fig. 9). The rate of meltwater contributions were less rapid on SHEBA and the pond storage increased somewhat more dramatically, perhaps because of the difference in ice type. The difference may be accounted for by the relatively high internal storage in FYI (Figure 10), which was not present during SHEBA.

## 4.2 Variability of the sea ice freshwater balance

There is variability in the sources and sinks in the sea ice freshwater budget both within our observational dataset (local scale), as well as across years and locations (basin scale). Here we consider the possible errors in our estimates that may impact the results shown and discuss the variability of relevant terms that may result in a larger range of values on the basin scale. In general, we expect the fractional distribution of sinks to vary more than sources due to the significant role of atmospheric and ocean forcing on small-scale heterogeneity, and these sink estimates also have more uncertainty in our calculations.

Variability in estimated source terms for this case study may result from snow thickness (which largely determines the snow melt input), sea ice bottom and surface melt, ridge fraction and thickness ratio, lateral melt rate, sea ice salinity, and liquid precipitation rate. Transect observations gave pre-melt (May) snow thickness of 31±17 cm (Itkin et al., 2023), which suggests up to almost 50 % variability in the local snow melt contribution. Similarly, sea ice bottom and surface melt rates vary significantly by thickness and ice type across the floe (Raphael et al., 2024), but the relative agreement across methods suggests the average contributions have errors less than 10 % (Figure 3). Sea ice core salinity values ranged from 1–5 g kg$^{-1}$ (Angelopoulos et al., 2022; Salganik et al., 2023a). Sea ice and density used in the volume conversions may present another possible source of error (Salganik et al., 2024), but are estimated to be small compared to other listed terms. Error and variability in lateral melt rate cannot be estimated from the observations available, but the relative magnitude is likely to be most sensitive to the size of the floe considered in budget calculations. Precipitation estimates may have substantial uncertainty, as described (Fig. 5), but should not vary spatially across the floe considered.

While the comparison with SHEBA suggests that there may be reasonable consistency in source terms across years and locations (Fig. 12), we would expect that there is significant basin scale variability in the magnitude and fraction of sea ice-derived freshwater. As snow on sea ice mostly melts by the end of Arctic summer, the snow melt term is mostly bounded by the seasonal maximum snow thickness, which has been estimated pan-Arctic as 20±6 cm from ICESat-2 and CryoSat-2 altimeters (Kacimi and Kwok, 2022). Variability in bottom and surface melt is at least in part determined by the ice age, which ranges from pure FYI to pure MYI across the Arctic. Coupled sea ice models may be used to explore the relative contributions

from mass budget terms in the context of the freshwater budget (e.g., Keen et al., 2021), where the changes in proportions of ice types likely lead to changes in dominant melt terms. One contribution to this is the range in ridge fraction, which can be estimated as varying from 12–20 %, as discussed in the methods. Finally, pre-melt sea-ice bulk salinity similarly varies from 1–2 $\mathrm{g\,kg^{-1}}$ for pure SYI to 4–5 $\mathrm{g\,kg^{-1}}$ for pure FYI (Vancoppenolle et al., 2009b), which may impact the conversion from sea ice meltwater to freshwater by up to 10 %.

Variability in sink terms may be a result of uncertainty in melt pond depth and fraction, melt pond salinity, depth and salinity of under-ice meltwater layers, depth of lead meltwater layers, lead or open water fraction, and variability in terms used in internal storage estimates, including meltwater salinity, ocean reference salinity, ocean heat flux, and initial snow thickness. In general, the uncertainty in each of these terms can be expected to be large. Melt pond volumes calculated using aerial and satellite photogrammetry suggest that the transect data used here may underestimate the equivalent freshwater thickness

by up to 50 % early in the time series (June 30) due to location bias, but estimates converge by mid-July because of pond drainage (Fuchs et al., 2024). Comparison of different methods for estimating under-ice and lead layer freshwater equivalent gives estimates that vary by as much as 100 % and 50 % on a given date, respectively. Direct estimates of the uncertainty in internal storage using ranges of observed input terms (Fig. 10) give an error around 10 % for the FYI core data, but estimates vary largely by ice type and model. Thus, the ratio of sink terms are likely to have significant error and variability spatially.

As a comprehensive budget of sea ice freshwater sinks has not previously been completed, we can not evaluate how this might vary. Melt ponds can vary widely in depth and coverage based on ice type, morphology, and forcing conditions (e.g., Buckley et al., 2020; Fetterer and Untersteiner, 1998; Webster et al., 2015, 2022). Freshwater layers under ice and in leads have been observed previously, but are also commonly not present in observations made across the Arctic during the melt season (Smith et al., 2023). We suggest that future work addressing this gap is needed. Modeling studies using coupled global climate

models may be well suited to address some of these questions regarding variability in meltwater storage in ponds and in the ice.

## 4.3   Comparison with a model: CESM2

Model representation of freshwater budget source terms from snow and sea ice meltwater and the storage in ponds is evaluated using one CMIP6 class coupled sea ice model, the Community Earth System Model 2 (CESM2; Danabasoglu et al., 2020).

Model outputs are evaluated using a sea ice grid cell near the MOSAiC location, where averages over 10 years from 2015–2025 are computed (Figure 13). The modeled total freshwater input from sea ice and snow melt (Fig. 13) is moderately low compared to that observed (Fig. 7), with about 0.35 m (40 %) less total freshwater input from sea ice and snow terms. There are more significant differences when comparing individual source terms. The model predicts more bottom melt but less surface melt compared to observations. Specifically, by July 25 there is approximately 0.1 m (50 %) more freshwater contributed from

bottom melt in the model compared to the observations, while the contribution from surface melt is about 0.35 m more (around 3x) in observations compared to the model. The snow melt and lateral ice melt are both lower in the observations than in the model. We note that MOSAiC provides only one realization of the range of possible conditions represented by the model, so

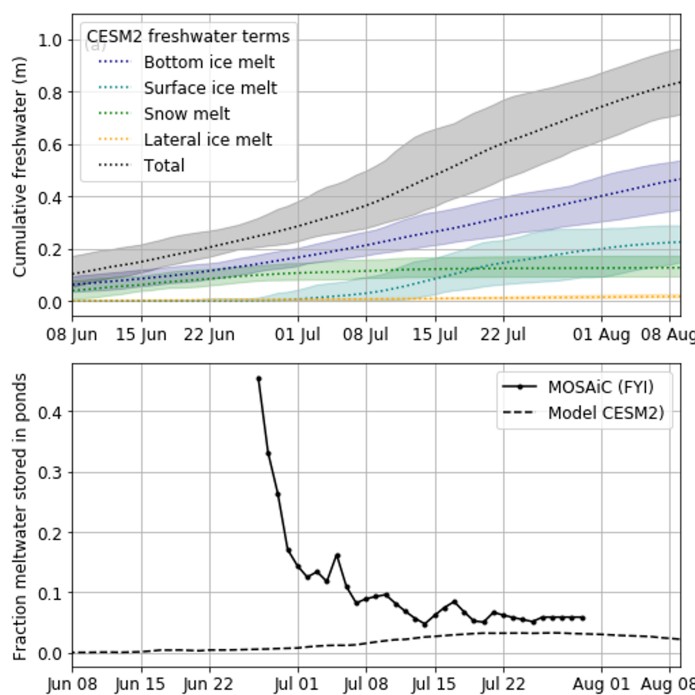

**Figure 13.** Sea ice freshwater budget terms from the CESM2 model near the MOSAiC expedition. (a) Freshwater source terms from sea ice and snow melt, with the range over 2015-2025 in CESM2 indicated by shading. (b) Fractional surface freshwater stored in ponds on MOSAiC (solid) compared to model results from CESM2 over the period from 2015–2025 (dashed).

future investigations with a forced 1D model will be necessary to understand the role of atmospheric or oceanic drivers versus processes in contributing to these differences.

Properly representing the storage of melt ponds in coupled climate models is critical to the representation of a number of other processes. We evaluate the representation of melt pond fractional storage in the model by calculating the fractional storage in the same manner as for observations (Fig. 13b). The model dramatically underestimates the fractional storage of surface meltwater in melt ponds, despite the under-representation of surface meltwater budget terms (Fig. 13a). The modeled storage never exceeds 4 %, and peaks much later than in the observations. Additionally, the greater estimates of melt pond volume

over the entire CO2 floe from orthomosaics early in the observational period (see Section 3.2; Fuchs et al., 2024) suggest that the observational estimates of pond storage (Figure 9b; 13b) may be biased low, which would push the observations and model estimates even further apart. Our working hypothesis is that this is a result of vertical meltwater drainage parameterizations which result in ponds that are too thin (e.g., Webster et al., 2022). Separate ongoing work is seeking to better understand the contribution of melt pond drainage processes to pond volume and improve the representation of the meltwater budget informed

by MOSAiC observations.

### 4.4 Impact on ocean heat budget

The presence of the observed thin, relatively fresh layers can have contrasting impacts on the upper ocean heat budget driven by local solar input. On the one hand, these surface layers may accumulate solar heat which can contribute directly to enhanced lateral and basal melting (Richter-Menge et al., 2001; Smith et al., 2023). On the other hand, solar flux can also be transmitted through such thin layers and the strongly stratified surface layer can allow solar heat to temporarily be stored deeper in the water column, inaccessible for immediate ice melt (Hudson et al., 2013; Granskog et al., 2015). Taken together, these opposing impacts suggest that the sea ice freshwater (salt) budget is closely intertwined with the upper ocean heat budget. Small-scale processes within the mixed layer are not typically observed by oceanographic observations nor represented in ocean circulation models (e.g., Steiner et al., 2004).

While we do not attempt a heat budget here, observations of upper ocean temperature combined with rates of melting suggest some feedbacks between the two. In the ice-covered portion of the study area, transmitted energy is likely directly available for sea ice bottom melt on the order of centimeters (Tao et al., 2024). In the open water portion (leads), the solar heating can be directly available for lateral melt. The heat available for melting is determined as a function of the temperature difference of the water from freezing, following Richter-Menge et al. (2001):

$$Q = \rho_{fw} c_{fw} (T - T_f) \Delta z \tag{7}$$

where $\rho_{fw}$ is the freshwater density, $c_{fw}$ is the specific heat capacity of freshwater, 4185 J kg$^{-1}$ K$^{-1}$, $\Delta z$ is the depth range of the layer, and $T - T_f$ is the difference between the water temperature and the salinity-determined freezing point. This is calculated here over the upper 2 m ($\Delta z = 2$) of leads using near-surface ocean profiles made on six dates in July (Smith et al., 2021a), shown in Figure 14. Salinity-determined freezing point is calculated using the Gibbs Seawater implementation of the Thermodynamic Equations of Seawater 2010 (TEOS-10) (McDougall and Barker, 2011). The lower salinity of the layer results in higher relative heat content for the same temperature. While there are lower-salinity layers present near the surface on all dates (Fig. 14b), the temperature above freezing is highest at above 2°C on 1 July and typically below 1°C after that (Fig. 14a). This results in a high heat content initially, which may have contributed to the initial increase in lateral melt rates (e.g., Fig. 7), which then stays at a relatively constant level for the remainder of the observation period.

The net impact of the freshwater budget on the heat budget cannot be determined, especially as the total solar heat flux is larger than the ocean heat flux from lateral and basal melt (e.g., Hudson et al., 2013). Regardless, observations suggest that the interaction of the two can increase lead heat content and lateral melt rates, which may persist throughout the melt season once initiated.

### 4.5 Meltwater composition from isotopic analysis

The oxygen isotope ratio and salinity of seawater may be used to differentiate sources as meteoric water (river runoff and precipitation) from sea ice meltwater (e.g., Macdonald et al., 1995). Isotopic analysis ($\delta^{18}O$) was undertaken as part of the

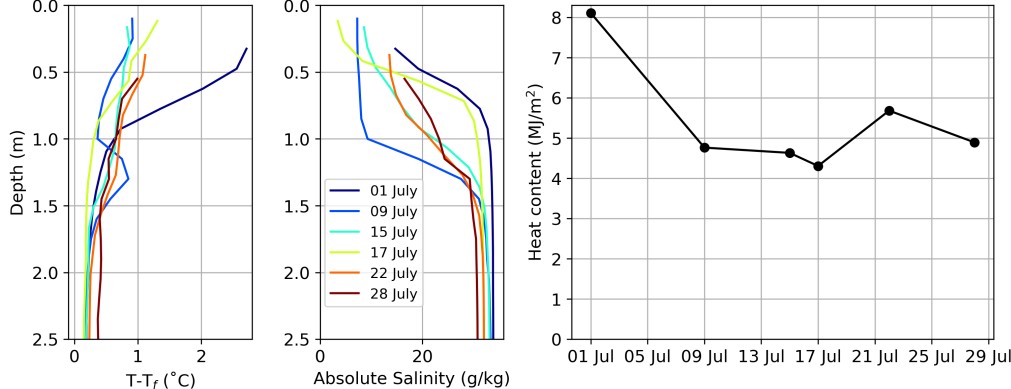

**Figure 14.** Upper ocean lead profiles of (a) temperature above freezing and (b) absolute salinity on 6 dates in July measured using Castaway CTD. (b) Lead heat content calculated by Eq. 7 on those 6 dates.

MOSAiC expedition to understand freshwater sources, and results relevant to the present manuscript can be found in Smith et al. (2022b); Mellat et al. (2024). Smith et al. (2022b) reported that melt ponds observed on 25 July 2020 were a majority FYI melt, with a significant secondary contribution from snow melt (Lange et al., 2022). The additional melt pond samples reported in Mellat et al. (2024) unfortunately did not have corresponding salinities recorded, so a composition analysis is not possible. Under-ice meltwater layer composition largely matched that of melt ponds (Smith et al., 2022b), consistent with the story that they are driven primarily by vertical drainage, rather than bottom melt. Under-ice meltwater layers were 31 % seawater, on average, suggesting some mixing between the meltwater layers and the underlying ocean by the sampling date on 25 July. Relatedly, the incorporation of snow and sea ice meltwater into the upper ocean (e.g., $V_{uo}$) contributes to significant changes in salinity and isotopic composition of surface seawater during the summer (Mellat et al., 2024). We also note that FYI has a relatively high enrichment of $\delta^{18}O$ compared to SYI. This difference is likely due to modification from the longer freeze-melt history and the lower values in surface seawater (Mellat et al., 2024).

## 5   Conclusions

This study builds on the results of Perovich et al. (2021), which examined meltwater budgets in the Western Arctic about 20 years prior (SHEBA), to provide the first comprehensive freshwater budget of FYI and SYI from observations, and quantification of the sinks. In general, there has been less deformed ice in the Arctic since 2007 (Sumata et al., 2023). As such, the substantially deformed floe measured during MOSAiC is not necessarily representative of the meltwater budget and partitioning of Arctic sea ice as a whole. However, comparison to meltwater budgets from SHEBA shows overall remarkable similarity between the source terms and melt pond storage. While these campaigns still only represent two discrete points in space and time, they can more robustly suggest some likely targets for model representation improvement, such as the storage of surface meltwater in melt ponds.

The largest source of local freshwater in the MOSAiC observations is from surface sea ice melt. Tao et al. (2024) found that a majority of solar energy input over sea ice during MOSAiC is absorbed within the ice, snow, and melt ponds, rather than transmitted through to the ocean, likely contributing to accelerated surface melt rates. This suggests the potential for feedbacks between the production of freshwater (as from melt) and storage in melt ponds, which modify the solar radiative budget. In comparison, bottom melt of sea ice constituted only around a third of cumulative freshwater in observations, but was notably increased by over a third by making adjustments for ridge fraction, which are underrepresented by typical observational methods. Precipitation sources in general are a small contribution to the freshwater budget (<10 % total), and the contribution from precipitation over the year stored as snow is much greater than that occurring directly during the summer as rain. As the Arctic transitions to be more rain-dominated in the future (Bintanja and Andry, 2017), the cumulative summer freshwater generation from snow will likely decrease, with a larger role for in situ precipitation. Lateral melt is a small freshwater source term on the relatively large floe observed here, as is typical within the pack ice, but is likely to be larger a contribution in marginal ice zones where typical floe sizes are smaller (e.g., Smith et al., 2022a).

The small total volume of freshwater sinks on, in, and under sea ice relative to the sources suggests that most freshwater ends up in the upper ocean. Even if we assume that errors in sink terms could be as large as 100 %, the ocean would remain at least the largest freshwater sink term, if not still the majority of the volume. More direct quantification of the upper ocean entrainment term in future studies would be useful for understanding the contribution to ocean freshening. Regardless, the storage of freshwater from sea ice and snow melt has significant local impacts. Despite their low volume, meltwater layers in ponds and in the ocean under the ice and in leads have extensive importance for the coupled Arctic system (Smith et al., 2023), acting as a barrier on the upper ocean, resulting in reduced gas and momentum exchange, separation of ecosystems, and more. Understanding the factors in formation and dissolution of these layers is key to understanding the importance across the changing Arctic. Observations from the MOSAiC expedition here show that while a higher fraction of surface freshwater is initially retained in melt ponds on FYI, SYI stores a higher fraction of freshwater through the mid-to-late summer. This is likely in part a result of the evolution of fractional coverage of melt ponds. Recent research has similarly suggested that melt pond fraction is initially higher on less deformed ice, but remains higher later in the summer on more deformed ice (Niehaus, et al., in prep), likely due to a reduction both in lateral drainage, as a result of the surface expression of deformation, and vertical drainage, related to reduced permeability with lower salinities. However, contradictory results throughout the literature suggest that this could be an important topic to address with future research (Polashenski et al., 2012; Webster et al., 2015; Buckley et al., 2020; Webster et al., 2022). An additional novel result of this study is that internal storage of freshwater in the sea ice through the desalinization process is larger than other quantified sinks. The associated brine drainage to the ocean results in a salinification of the upper ocean at the same time that melt is directly freshening, making observational constraint of the freshwater budget in the ocean even more challenging.

*Data availability.* All MOSAiC datasets used have been archived and are available at the following references:

– Stakes data: Raphael et al. (2022)

– IMB datasets: Several datasets were used in this analysis, including SIMB archived at Perovich et al. (2022), DTCs archived at Salganik et al. (2023), and SIMBAs which are individually archived at the following DOIs: 10.1594/PANGAEA.940393 (T58), 10.1594/PANGAEA.940231 (T62), 10.1594/PANGAEA.940593 (T63), 10.1594/PANGAEA.940617 (T64), 10.1594/PANGAEA.938134 (T66), 10.1594/PANGAEA.938128 (T67), 10.1594/PANGAEA.940659 (T70), 10.1594/PANGAEA.940692 (T74), 10.1594/PANGAEA.940740 (T75), 10.1594/PANGAEA.940702 (T76), 10.1594/PANGAEA.940712 (T79)

– Floe orthomosaics: Neckel et al. (2023)

    – Snow freeboard from airborne laser scanner: Hutter et al. (2023)

    – Precipitation datasets: Kyrouac and Holdridge (2019)

    – Melt ponds: Itkin et al. (2021)

    – Melt pond salinities: Lange et al. (2022)

– YSI lead and under-ice: Smith et al. (2021a)

    – Fishing rod CTD: Karam et al. (2023)

    – Coring: Oggier et al. (2023a, b)

    – Upper ocean: Schulz et al. (2022)

    – SHEBA mass balance data: Perovich et al. (2007)

 **Appendix A: Open water fraction estimates**

Figure A1 summarizes the methods related to the calculation of open water fraction. The area over which open water fraction is calculated from orthomosaics (Neckel et al., 2023) is shown in the images in the top row, with red shaded rings. The sea ice-covered fraction, where open water fraction is one minus the sea ice fraction, is plotted in the bottom panel in red, which are interpolated to dates between. For context, they are compared with two satellite methods for estimated sea ice-covered fraction, which generally bracket the same range of values, giving higher confidence to the estimates.

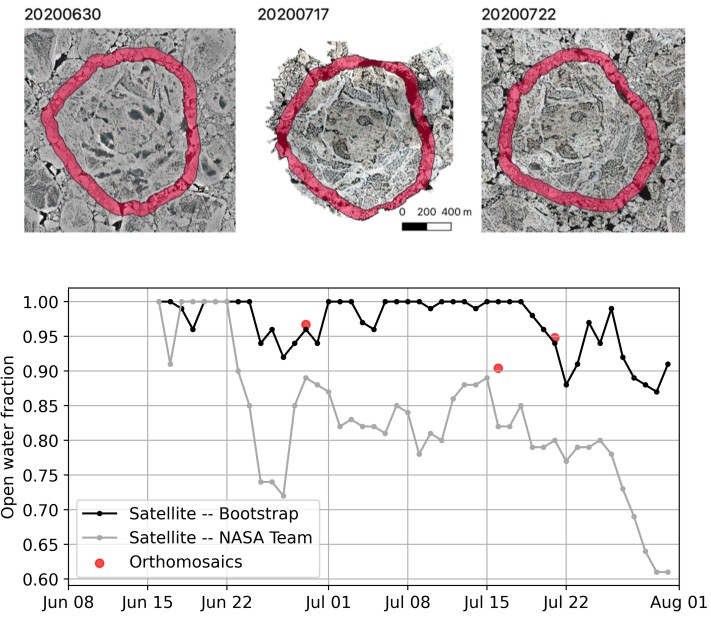

**Figure A1.** Methods for calculation of open water fraction. Top row shows aerial orthomosaic maps (from Neckel et al., 2023) of floe area and the 100-m ring around the floe used to determine open water fraction on 3 dates. Bottom plot shows comparison of open water fraction (derived from top images; red circles) with satellite sea ice products (which have spatial resolution of 25 km x 25 km).

**Appendix B: Upper Ocean, $V_{uo}$**

The fresh water content of the upper ocean $h_{FW}$ can be calculated from profiles of vertical salinity $S_A$ by integrating the deviation in salinity from a chosen reference salinity $S_{ref}$ over some vertical range $z_t$ to $z_b$

$$h_{FW} = \int_{z_t}^{z_b} \frac{S_{ref} - S}{S_{ref}} dz, \tag{B1}$$

following, e.g., Rabe et al. (2011). Here, we calculated $h_{FW}$ from daily averaged salinity profiles obtained with a microstructure profiler at a central position of the MOSAiC floe (Ocean City, Schulz et al., 2022). Several profiles were measured on a near-daily basis through a hole in the sea ice between 27 June and 29 July, 2020, no data is available on 12, 23 and 24 July 2020. In contrast to hydrographic measurements performed from the ship, which are affected by mixing created at the ship's keel and by its positioning system, the microstructure measurements represent a more undisturbed upper ocean stratification. Vertical

integration was chosen to start at $z_t = 4$ m to avoid including salinity measurements in a fresh water lens that was present at the sampling site until mid June 2020. The lower limit was either chosen as a fixed depth level (50 m) or a fixed isohaline (32.4 g kg$^{-1}$), the reference salinity was chosen to be 34.2 g kg$^{-1}$. Results are insensitive to the choice of the lower integration threshold criterion and variations of the exact depth level or isohaline, except towards end of July (see Fig B1b), when the 34.2 g kg$^{-1}$ isohaline had deepened from initially between 4–60 m to over 100 m. Results are quantitatively more sensitive to

the choice of reference salinity, however, the qualitative behaviour of $h_{FW}$ holds for a broader variation of $S_{ref}$.

Ignoring the presence of lateral gradients or advective effects, the evolution of $h_{FW}$ would reflect the input of fresh water into the upper ocean. However, the drift traversed substantial lateral salinity gradients associated with entering the edge of the transpolar drift of river-rich water on 16 July, 2020 when leaving the Yermak Plateau, and the subsequent progression further into the low salinity core (vertical dashed line, Fig. B1a). This transition into a different surface water regime is also visible

in other parameters than salinity, i.e., an increase in the river water fraction (based on oxygen isotope measurements) and a higher concentration of Colored Dissolved Organic Matter characteristic of river water (see Schulz et al., 2024, in press). The evolution of $h_{FW}$ after 16 July, 2020 is therefore dominated by regional gradients in surface salinity, and does not provide means to discern local meltwater input into the upper ocean.

Between 27 June and 16 July, 2020, we did not observe any changes in upper ocean properties that point to pronounced lateral

gradients. For this period of time, we can assume the variability in $h_{FW}$ reflecting of meltwater input, acknowledging that this involves the assumption of no lateral effects, which cannot be proven. Ocean freshwater content exhibited little variability until July 11, with values ranging between 0.1–0.2 m before 6 July, 2020 and 0.2–0.3 m after. Sometime between 11–13 July, freshwater content increased by around 0.35 m to 0.6 m, likely associated with the substantial melt pond drainage event Webster et al. (2022), as well as vertical mixing events which redistributed melt water from a shallow, meter-scale layer near

the surface through the upper tens of meters in the ocean. A subsequent slight decrease in $h_{FW}$ of 0.1 m over three days could be attributed to a lateral spreading of the localized meltwater input driven by baroclinic instabilities.

Overall, we conclude that in our case the upper ocean freshwater content by itself is not suited to quantify meltwater input into the ocean. The contribution of spatial gradients or lateral advection cannot be isolated from the meltwater input, and might dominate the signal, especially near hydrographic fronts.

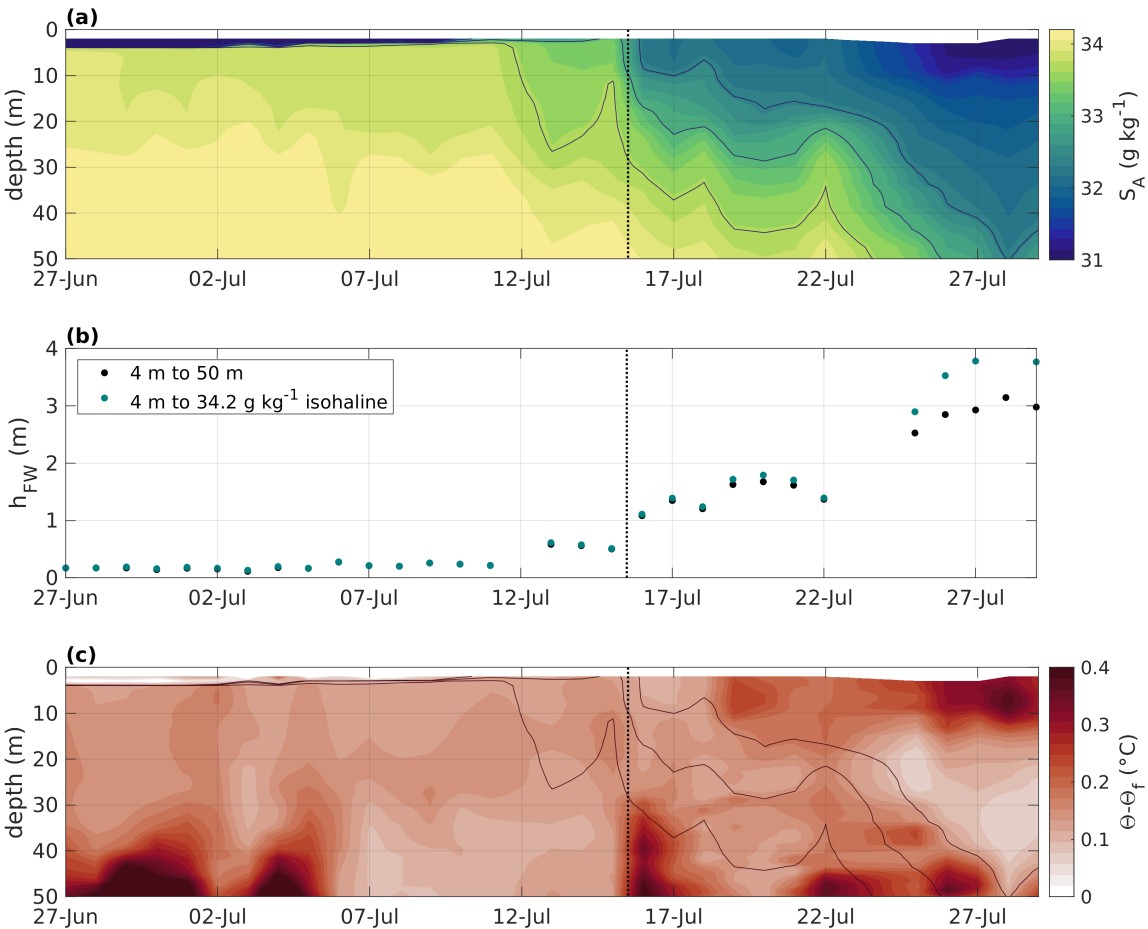

**Figure B1.** Time series of (a) upper ocean Absolute Salinity (g kg$^{-1}$), (b) fresh water content (m), and (c) conservative temperature $\Theta$ deviation from freezing temperature $\Theta_f$ (°C). Vertical dashed lines indicate the timing of the front between river-water poor and rich surface regimes.

**Appendix C: Internal Storage**

Figure C1 compares vertical profiles of salinity estimates from IMB buoys in FYI and SYI (left, middle panels) and the resulting change in salt mass over time (right panel). The salt mass, $m_S = S_s i \cdot h_s i \cdot rho_s i$, decreases from the start of the melt season onwards. Figure C2 then compares bulk brine volume estimates from coring measurements (circles) and from IMB buoys nearby coring sites (dashed lines). The relative volume of brine was estimated using ice salinity and temperature measurements (Figure C3) from ice coring using Cox and Weeks (1983) for cold ice and Leppäranta and Manninen (1988) for ice warmer than -2°C. We note that the bulk ice temperature is notably warmer for FYI in much of July (Fig. C3) due to the presence of thin meltwater layers, which also impacts the simulated brine volume. To fill the observational gap between 4 May and 22 June, brine volume was also estimated using temperature measurements from two IMB located near coring FYI and SYI sites, with linearly interpolated salinity from ice coring. The close agreement in magnitude and approximate timing suggests that coring observations provide a reasonable estimate of salt mass flux from the ice on both FYI and SYI.

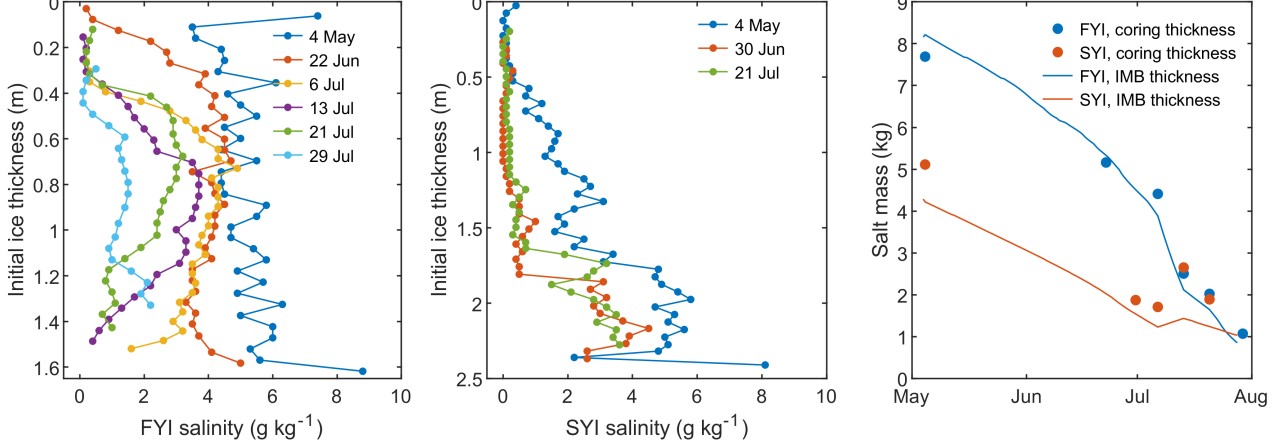

**Figure C1.** Vertical profiles of first- (a) and second-year ice salinity (b), and temporal evolution of salt mass of first- and second-year ice

Bulk ridge salinities (Figure C4) show general freshening trends of ridges over time. These generally follow the freshening trends of level FYI and SYI (e.g., Fig. C1), and support the approach of estimating combined internal meltwater storage.

*Author contributions.* MMS conceptualized the study. MMS, NF, DKP, IR, ES, MAG, KS, MDS, and MW curated data for analysis, and MMS, ES, NF, KS contributed to formal analysis. MMS, NF, and KS contributed to visualization. MMS led writing of the draft, with all authors contributing and approving the final draft. Funding acquisition: MMS, MAG, KS.

*Competing interests.* The authors declare that no competing interests are present.

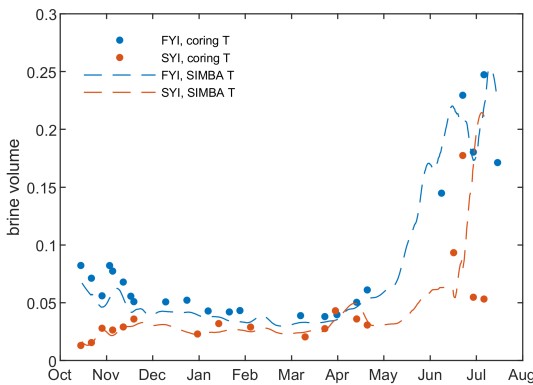

**Figure C2.** Bulk brine volume for level FYI (blue) and SYI (red) from cores (circles) and IMB buoys (dashed lines).

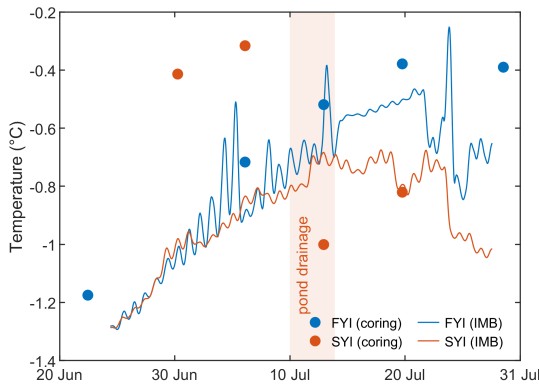

**Figure C3.** Bulk ice temperature for level FYI (blue) and SYI (red) from cores (circles) and IMBs (lines).

*Acknowledgements.* MMS was supported by NSF OPP 2138787. ES and MAG were supported by the Research Council of Norway project HAVOC (grant no 280292) and ES through project INTERAAC (grant no. 328957). MAG acknowledges support from Hanse-Wissenschaftskolleg Institute of Advanced Study (Delmenhorst, Germany) and funding from the European Union's Horizon 2020 research and innovation programme under grant no. 101003826 via project CRiceS (Climate Relevant interactions and feedbacks: the key role of sea ice and Snow in the polar and global climate system). KS received support from the National Science Foundation, grant NSF-OCE-2401413. NF acknowledges funding from the BMBF project NiceLABpro (03F0867A) and from the Deutsche Forschungsgemeinschaft under Germany's Excellence Strategy (EXC 2037; CLICCS – Climate, Climatic Change, and Society; project no. 390683824). MW conducted this work under the NSF Project 2325430 and NASA's Interdisciplinary Research in Earth Science project 80NSSC21K0264. MDS was supported by the U.S. Department of Energy (DOE, DE-SC0021341), and the NOAA Physical Sciences Laboratory (NA22OAR4320151) and Global Ocean Monitoring and Observing Program (FundRef https://doi.org/10.13039/100018302). IAR was supported by by NSF OPP-1724540 and NSF OPP-1724424.

Data used in this manuscript were produced as part of the international Multidisciplinary drifting Observatory for the Study of the Arctic Climate (MOSAiC) with the tag MOSAiC20192020 and the Project_ID: AWI_PS122_00. We thank all people involved in the expedition of

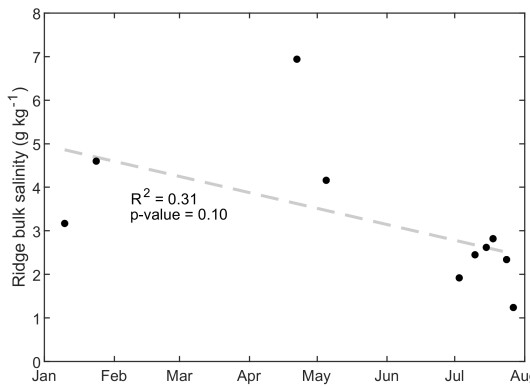

**Figure C4.** Bulk salinity of various ridges sampled within CO1.

the Research Vessel Polarstern Knust (2017) during MOSAiC in 2019–2020 as listed in Nixdorf et al. (2021). Precipitation data was obtained from the Atmospheric Radiation Measurement (ARM) User Facility, a DOE Office of Science User Facility Managed by the Biological and Environmental Research Program. We acknowledge Dave Bailey and Marika Holland for assistance with CESM2 outputs. The CESM project is supported primarily by the National Science Foundation. Computing and data storage resources, including the Cheyenne supercomputer (https://doi.org/10.5065/D6RX99HX), were provided by the Computational and Information Systems Laboratory (CISL) at NCAR.

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
