# Peer review of "Formation and fate of freshwater on an ice floe in the Central Arctic"

_EGUsphere, 2024_

## Author Response (AR1)

Author response to RC1, Cathleen Geiger:

Overall this is a well-done paper. It is clearly written with great effort taken to actively measure those terms often used in numerical models, but to do so through great effort by going into the field and directly measuring as many inputs as possible and then compute the actual results using a box model approach which is then compared to a current and well respected community model. These types of papers are significant because they are the ground truth which reveals where the physics in the models need to focus so as to advance closer to reality. This paper clearly involved a ton of team work- which has been carefully done and so I strongly recommend that this paper be accepted for publication pending one notable recommendation to improve the paper further.

One highly recommended extra step to significantly improve the paper is the following. When one first steps onto snow-covered sea ice, the first thing that becomes crystal clear is the variability of absolutely everything around you. Take one step and you are able to measure a whole new set of values. A simple way to bring all that rich variability into a paper like this is to create a table of each type of measurement and then explore the impact of propagated variability. Specifically, which terms in the equations will yield the highest variability in the results? In other words, which of the many collected measurements are the most influential in widening the outcome of the results? As an exercise, I recommend that the authors create a table of each measured variable and then estimate either in absolute or percent-relative terms the degree/range of variability observed within the red perimeter study area.  As example, the lateral melt range is discussed as being large but negligible to freshwater estimates. That's a good start, but that discusses away a source as negligible. What are some of the variables that have a large range but are not negligible and which of these may vary so as to impact the overall conclusion of the paper? One does not need to do an exhaustive study, rather pick judiciously those variables which have the greatest weighting factor on the outcome.

Take for example, Figure 3- cumulative freshwater (m). The Surface melt is a bigger producer than the bottom and the bottom a bigger producer than the snow. But how much do equations (4), (5), and (6) change if the ice salinity is not fixed at 3 g kg^-1 (line 90) and on line 254 the salinity of the surface meltwater is not constant at 1.1 g kg^-1. I don't find this constant in the Lange et al 2022 paper- rather I find a range of values in the Lange et al 2022 paper as an excellent source from which to gather variability statistics. What happens to the results if the ice salinity ranges 2-4 g kg^-1 and the surface meltwater is not constant but  ranges 0.5-1.5 g kg^-1 ? These are small perturbations and easily possible within a few horizontal meters. Do small perturbations like this impact the outcome ?

Same goes for the 1 W m^-2 ocean heat flux. What happens if the heat flux is in a range of 0.5 to 3 W m^-2 and the snow variability is 10-30 cm instead of a constant 22 cm?

Same with ridge macroporosity. In early June before solstice, it may still be 4-6% but the freshwater equivalent time series goes into August. Is 4-6% still valid for ridge macroporosity when melt ponds are experiencing bottom drainage? Ridges are polar reefs of biodiversity in the Arctic in Summer. There is surely a wide range of salinities and porosity in a single ridge between start of June and start of August.

In particular, Equation (3) computes the internal meltwater volume as a residual difference between top and bottom. In that case, which of the two inputs is more responsible for the higher variability in V_internal- is it the top or the bottom and why? These are really effective sensitivity questions that can be answered by inputting a range of values instead of a constant number.

In particular, Figure 7 would benefit from some vertical whiskers especially just prior to the surface drainage event at the end of June.

Could variability in bottom melt from 15 to 22 June have an impact on the big surface ice freshwater drainage event between June 22 and July 1? If so, then which input(s) is/are responsible? I suspect that bottom and surface melt process are highly intertwined as the porosity in the microstructure increases toward 5% as the season progresses from June to August.

In short, this paper will elevate from a point measurement sample study to a representative regional study if the authors run some sensitivity tests to get a sense of ranges of outcomes in their freshwater volume calculations. And especially, if they identify which variables most impact the sensitivity of these calculations. Those are the variables that need to be prioritized in models to ensure that the most critical meltwater physics are nominally incorporated.

I suspect that the calculations are already in a code so it is the matter of making a loop around that code and running some realizations with a range of values instead of constant numbers.

My biggest concern with constants is that the constant will stick and it will be very hard to undo once done. It is unwise- especially now- to assume something constant when in fact variability itself is on the rise.

> *Thank you very much for the positive and useful review. The suggestion to include some quantification and discussion of sensitivity and error is well-taken. The possible sources of error and variability in each term of the freshwater budget were evaluated through creation of a table (ultimately not included in the revised manuscript) and carefully incorporated through the manuscript where possible.*
>
> *We have also added some error estimates to parameters and terms, where possible, and discussion of the range of likely values. These include:*

- *Precipitation. We now include error estimates for the two methods, which are also combined into an average error which has been propagated through.*
- *Internal storage. We now include sensitivity analysis to input terms, which are provided as error estimate (shading) on Figure 10. This includes varying ocean heat flux from 0.5-10 W/m$^2$, references salinity from 32-34.9 g/kg, and meltwater salinity from 0.6-1.9 g/kg.*

*The largest change in response to this review is the addition of subsection titled, "Variability of the sea ice freshwater budget". This discusses both uncertainties within our dataset, as well as likely sources and magnitudes of variability on the basin scale and interannually, for both sources and sinks. Please see the revised manuscript for the full text.*

*Additionally, we edited wording throughout to highlight that the exact ratios are likely to be different in different years and locations, e.g., by adding "...once instance of the freshwater budget..." to the abstract. The conclusions highlight the biggest sources of uncertainty and variability.*

Author response to RC2:

Melting of snow and sea ice produce relatively fresh water, adding into the upper ocean, and further impact the interactions among air, ice and ocean. This physical process is complicated, and some important quantitative fractions within which are still unknown. Based on the field investigations during MOSAiC, the manuscript presents a detailed anysis on the sea ice freshwater budget, and the fate of meltwater over time. The manuscript is well organized and easy to follow. I have only some specific comments as below.

*Thank you for your review of the manuscript and positive feedback. Specific comments have been incorporated into our revision, which are described in italics below.*

1. In the first paragraph of introduction section, I think the importance of the sea ice meltwater should be clarified more clearly rather than only saying "The magnitude and fate of freshwater associated with sea ice melt are important for the surface energy budget, ice mass balance, ocean structure and primary productivity" on lines 24-25. The relevant description can be more specific.

   *The specifics of the impact of the magnitude and fate of freshwater budget are included in paragraphs at lines 34-58. Some additional detail has been added to these paragraphs, which can be seen in the tracked changes version.*

2. There are some parameters I do not quite understand. For example, why do you chose 3 g kg$^{-1}$ in line 90? The salinity of sea ice surface is generally low, and the same to pond bottom ice.

   *As the reviewer states, the summer sea ice salinities are generally quite low, and this value is within observed ranges. We have added some additional text for context (**bold** text is new): "The ice salinity is taken as a fixed 3 g kg$^{-1}$, which falls within the observational range for both ice types **and is between the values of 2 ppt determined to be characteristic for Arctic summer sea ice in Vancopenolle et al. (2009) and the assumed value of 4 ppt in Aagaard and Carmack (1989)**."*

3. The curves in the figure can be more differentiated. For example, figure 7. You can use both dash and dot lines to highlight the difference.

   *Dash and dot lines are used elsewhere to delineate different ice types (e.g., Figure 9, 10) or experiments/data sources (Figure 12, 13), so we avoid using here as well. I am guessing that the reviewer is having the most trouble differentiating surface and snow melt (teal and green), so color of snow melt is changed to a lighter and more yellow green to differentiate.*

4. When compared figure 7 with figure 13, more quantitative descriptions should be added. On lines 377-378, "the modeled total freshwater input from sea ice and snow melt (Fig. 13) is moderately low compared to that observed (Fig. 7)". How much and why?

> *Edits have been made to the rest of the referenced paragraph to make the comparisons between the source terms in observations and the model more quantitative and specific. This includes the addition of "…with about 0.35 m (40 %) less total freshwater…" to the sentence referenced in the comment, as well as the addition of "by July 25 there is approximately 0.1 m (50 %) more freshwater contributed from bottom melt in the model compared to the observations, while the contribution from surface melt is about 0.35 m more (around 3x) in observations compared to the model". As the last sentence of the paragraph states, we cannot be more specific about why these differences may be without dedicated modeling sensitivity studies.*

5. On lines 391-392, "hence the underestimation from the model even more dramatic than presented here". Can a clearer explanation be given? The comparison between obervations and model results shows a big difference. Is there any possiblity to improve the parameterization of freshwater budget in the numerical modling accroding to these observations?

> *Yes, a hypothesis for the model-observational disagreement and a path towards a solution are discussed in the two sentences following that referenced. We have edited the paragraph slightly to increase clarity and specificity.*

6. What is the perpose of section 4.3? As you say "The net impact of the freshwater budget on the heat budget cannot be determined" on line 417.

7. Lines 396-406, I think these are not discussions.

> *Both this and the comment (6) above refer to the section "Impact on ocean heat budget". As the data to complete a full heat budget is not available, we do not include it in the main results section. However, we argue that there is evidence from prior work that the meltwater budget has important implications for the heat budget, and thus it is important to discuss. Some small edits are made to this section to clarify the intent, but we have opted to retain it for in the present location to contribute to ongoing discussions about the relationship of freshwater and ocean heat budgets.*

---

## Author Response (AR2)

Author response to RC1, Cathleen Geiger:

The paper is ready for publication, but I invite the authors to please include more quantitative information in the abstract beyond the simple percentages. In checklist item #3 above, I can only rate the significance as Fair per the definition given where the paper stands now- simply because there are few new practical applications of broad relevance given how the outcome is communicated. Therefore as my final recommendation, please consider a significant upgrade to the Abstract- which is the key to this paper being read. And if you do, then I can say that it elevates my ranking in #3 from Fair to Excellent.

Keyword missing in the abstract is Process. It's in the introduction and all over the paper, but not in the Abstract. A field experiment like this is looking at singular events to gain insight into geophysical Processes. Please include this word in the Abstract somewhere near the reference to the words "one instance". It is incredibly important in a field paper such as this to explain why this paper is relevant in the larger context. The word "Process" is used as a keyword search by modelers to find papers on this subject- especially process modelers. Contrary to this, the words "one instance" makes it sound anecdotal- which takes away from the significant merits of the paper. You want the chance to reproduce such activities again- say in 2032 IPY- yes?.

For field people to continue going to the field, it is crucially important to clearly spell out WHY IS THIS STUDY IMPORTANT IN THE LARGER CONTEXT- in the Abstract. Clarifying this as a "Process study of the formation of fate of freshwater on ice floes"- in the Abstract- brings such a context. The jump in the abstract from single "instance" to CESM2 climate model currently reads as too big to grasp in the Abstract. The hook is missing in the abstract to make people want to read the paper.

As example, percentages are a fuzzy thing to compare to a model, but a rate of freshwater production or rate of drainage can be inferred and included in the Abstract? Rates can be computed in process models as a next step before jumping into the big climate models. I couldn't find any mention of the encouragement of process models to help scale these results up toward climate models. But maybe I missed that. If it's in the main body, it helps to bring that to light in the abstract.

I am essentially challenging you to include something tangible AND TESTABLE directly in the abstract- something that modelers can sink their codes into to see if they are on track. As example, in the last paragraph- "dramatically underestimated"- please put a number with "the drama" so that we can also see how big that underestimate really is and what model-derived term is responsible- like freshwater production rates as wonderfully highlighted in Figure 7. That

is a notable result that can be tested first against a process model and then upscaled to a climate model.

As example, what is that rate - on average - in terms of Meters Per Summer? Granted, this is not SI units, but it is understandable in terms of impact. A freshwater production rate of 0.01 to 0.02 m/day (10-20 cm/day) times two months (~60 days), the ballpark is 0.5-1.0 meter per summer in freshwater production which is a seasonal average (that matches up with 7a and 7b). The rate of 0.5-1.0 meter / summer season is a climate rate that can be tested in a climate or process model that includes melt pond fraction and in turn be directly compared to this "one instance" field experiment. A number like this essentially provides a bulk term to compare with winter ice growth rates as climatological inventory numbers. In a model, this can be used to explain why a climate model may be underpredicting ("dramatically") the rate at which ice is being lost due to summer freshwater melt in excess of winter growth.

A quantitative statement like this let's you leverage modeling results the next 5 years to get you out there again in 2032 for IPY.

*We really appreciate this external perspective on how the abstract can best hook readers and motivate the work. As you can see from the track changes version, we have made heavy changes to the abstract in response to the comments. Notably, we have made a more concrete suggestion that the evaluated terms (such as cumulative summer freshwater) might be used as model diagnostics. The word "process" has been incorporated at multiple points throughout the abstract. We hope that these changes capture the intent of the reviewer's comment and provide more tangible results and relevance for future work to build on.*